

# Transient Flow Patterns of an Annular-like Stratospheric Polar Vortex.

Huw C. Davies[1], Michael A. Sprenger[1]

[1]Institute for Atmospheric and Climate Science, ETHZ, Zurich, 8092, Switzerland

*Correspondence to*: Michael A. Sprenger (michael.sprenger@env.ethz.ch)

**Abstract.** A two-component study is undertaken of flow features associated with the wintertime polar-night vortex at upper-stratospheric levels. First, three cursory case studies are presented based upon reanalysis data. They suggest that: - (a) sub-planetary scale flow features are pervasive, tend to occur near the periphery of the vortex, and are associated with the vortex's predilection to develop a structure akin to an annular-like band of enhanced potential vorticity; (b) planetary scale flow features are influenced by, and can significantly distort such a band. Second, theoretical considerations and numerical model simulations of perturbations of an annular band of enhanced absolute vorticity using a non-divergent barotropic model on a polar $\beta$-plane yield results consistent both with the occurrence of the sub-planetary scale features, and with the possible rapid reconfiguration of a pre-conditioned planetary scale flow. The latter result bears comparison with the occurrence of a sudden stratospheric warming event via the synergetic combination of strong planetary-scale Rossby-wave forcing from below acting upon the vortex's annular potential vorticity band.

## 1. INTRODUCTION.

The general form of wintertime stratospheric polar vortex (SPV) is well established (see e.g. Waugh and Polvani, 2010; Schoeberl and Newman, 2015; Butchart, 2022). It extends from the lower stratosphere at ~ 15km to well beyond the stratopause at ~ 50km, attains a maximum wind strength often greater than 120 ms$^{-1}$ at stratopause levels, and is characterized in the stratosphere by an inner cold core driven primarily by the lack of ozone-related radiative heating during the polar night.

On quasi-horizontal or isentropic surfaces, the SPV's lateral 'edge' has been defined in various ways (Waugh et al 2016; Manney et al 2021). Kinematically it has been associated with the iso-line demarking the strongest wind strength (Waugh et al 2016) or the maximum streamline density (Harvey et al 2002; Harvey et al 2009), cartographically as a feature-based configuration (Serra et al 2017), and dynamically as the zone of sharp lateral PV gradient (Butchart & Remsberg, 1986; Manney et al 1994; Nash et al 1996). The latter definition arises from regarding the vortex's lateral configuration as an inner core of high potential vorticity (PV) encased by a narrow zone of strong PV gradient that is itself encircled by broad and well-mixed surf zone of lower PV (McIntyre & Palmer, 1984). The adoption of one or another of the above criteria to define the vortex's periphery will depend upon its practical utility or its interpretative value for the specific purpose in hand.

The present study is geared to examining the planetary and finer-scaled flow features that are contiguous to, or related to, the structure of the SPV's periphery, and to this end we adopt a PV perspective (Hoskins et al, 1985). The study is framed around the postulate that a, possibly fragmented, annulus of enhanced PV (- i.e. a maximum in the across-jet distribution of



the PV) can exist near periphery. This assertion is rendered plausible by two factors. First, a strong laterally confined jet with a relative vorticity maximum (say, $\zeta_{max}$) located at a co-latitude $\chi$ ($< 30^0$) on the vortex's poleward side would connote a maximum of absolute vorticity provided $(\zeta_{max}/2\Omega) > \frac{1}{2}\chi^2$, and this inequality is readily satisfied (- often by an order of magnitude) in the upper stratosphere. This in turn would connote a PV maximum (*sic* annulus of enhanced PV) if the variation of the stratification on an isentropic surface within the vortex is not comparably large. Second, the presence of such an annular structure on an isentropic surface is a necessary criterion for instability of axisymmetric quasi-geostrophic stratospheric flow (Charney and Stern, 1962), and the realization of the instability could result in a fragmented annulus.

The probable existence of such an annular structure in the polar stratosphere has long been recognized (Hartmann, 1983), and indeed a balanced flow response to a radiative equilibrium (Leovy, 1964) or radiatively determined (Shine, 1987) stratospheric state would possess such a maximum. Such a feature has also been detected in other planetary atmospheres (Sharkey et al 2021; Shultis et al, 2025). Moreover, Davies and Spengler (2024) - hereafter denoted DS24 - drew attention to flow features at stratopause elevations that are akin to that of a fragmented annulus.

An annular structure of enhanced PV would be embedded within an SPV that itself exhibits a distinct life cycle (Lu et al 2021), and marked inter-annual variations (Butchart, 2022; Rao et al 2024) particularly those related to the occurrence and aftermath of a Sudden Stratospheric Warming (SSW). This poses questions as to whether the annular structure or its fragmented counterpart is present at all phases of the life cycle, and what role (if any) it plays in the development of SSW events (Baldwin et al 2021) that have themselves been pointedly linked to a variety of mechanisms and preconditioned states deemed to be more amenable to engendering such events (Plumb, 1981; Fredricksen, 1982; Smith, 1992; Matthewman and Esler, 2011; de la Camara, et al 2017; Lawrence and Manney, 2020; Yessimbet et al, 2022).

Again, the annulus occupies the same spatial domain as two classes of shorter-term phenomena, and questions arise regarding the extent of their inter-dependence. One class can be viewed as intrinsic natural modes governed by the structure of the SPV itself such as planetary-scale zonally propagating waves (Venne & Stanford, 1979; Randel & Lait, 1991), smaller scale zonally propagating waves and synoptic-scale features (Lu et al 2021). The second class comprise planetary-scale Rossby waves (PRWs) (see e.g. Hitchcock and Haynes, 2016), and short-term smaller-scale gravity waves (GW) (see e.g. Holton & Alexander, 2000; Eichinger et al 2020) that propagate upward from lower elevations and can break on attaining large amplitude. The PRWs tend to break in the mid-to-upper stratosphere and the GWs at even higher elevations, and on breaking can heavily modulate the SPV's structure. Indeed, breaking PRWs (Hitchman and Huesmann, 2007; Greer et al, 2013) strip PV filaments off the vortex edge, create the surf zone (McIntyre & Palmer, 1984), and are endemic to the occurrence of SSW events (Martius et al, 2009). Likewise breaking gravity waves can serve as a drag (GWD) reducing the strength of the SPV's jet.

In this study attention is focused on the presence, role and dynamics of a PV annulus and/or its fragmented counterpart in the wintertime upper stratosphere. It is set against the backcloth of the series of questions posed above, and it has observational and theoretical components. For the observational component, cursory illustrative case studies pertaining to the



SPV's annular structure are presented for three different winter settings. For each case attention is drawn to the structure, temporal evolution, and dynamical character of the realized transient sub-planetary and planetary scale flow features.

The theoretical component is prompted by noting that the observationally detected features (DS24) bear comparison with the break-up of a barotropic jet-like flow. The ready inference that an SPV's jet can satisfy a necessary condition for barotropic instability has spawned a fleet of studies for basic states akin to that of either the SPV's jet or the circumpolar jets

of other planetary atmospheres (see for example, Pfister, 1979; Davies, 1981; Hartmann, 1983; Michaelangeli et al., 1987; Manney et al., 1988; Seviour et al 2017; Mitchell et al., 2021; Waugh et al., 2023). The strategy adopted here is to select an idealized theoretical model capable of capturing the essence of rapidly growing wave patterns of a barotropic model for flow settings akin to that of a deep undisturbed SPV. The objective is not to replicate (or contrast) the results with that of the earlier studies but rather to seek further basic understanding of and insight upon the dynamics of the observed features.

Both components are foreshadowed by the recognition that the presence of an annular PV feature in the upper stratosphere would not only alter the SPV's structure itself but also influence the character of in-situ generated phenomena and of waves propagating into the domain. For example, a change in the SPV's structure modifies its transmissivity to upward propagating waves (Hitchcock and Haynes, 2016). The strong positive PV gradient on the outer side of the annulus constitutes a columnary shaft that favours for vertical PRW propagation (Abatzoglou and Magnusdottir, 2007), and contrariwise the

negative PV gradient on its inner side would serve not merely to inhibit but preclude propagation into the vortex core, and thereby in principle help isolate the core from external effects. It is pertinent to note that the 'shielding of the vortex core' associated with the PV gradients contrasts markedly with the dynamics at the counterpart tropopause-level sharp PV gradient (and accompanying jet stream) located at the break in the mid-latitude extratropical tropopause (Appenzeller and Davies, 1992).

The paper is arranged as follows. Section 2 sets out the observational case studies. Section 3 sets out the theoretical results derived using an idealized barotropic model to study the dynamics of both small and finite amplitude perturbations of an SPV-like basic state. In a final section consideration is given to the limitations and relevance of the study and in particular to examine the link between the observed flow features and the model results.

**2.0  Observed Flow Features.**

Three cursory case-studies are presented with the objective of capturing the essence of transient flow features associated with an annular-like PV structure in the wintertime upper stratosphere.

**2.1  Data.**

Consideration of the flow at stratopause and higher elevations using reanalysis data sets are subject to constraints arising from the decrease in the number and resolution of sampled meteorological variables and the shortcomings of the parent assimilation model at these elevations (- see the overview in S-RIP, 2022). Nevertheless, Reanalysis fields have been utilized, albeit with



caveats, to examine a range of upper stratospheric diagnostic studies. Moreover, DS24 showed examples of the ERA-Interim (Dee, et al 2011a,b), MERRA (Gelaro, 2017), and ERA-5 (Hersbach, 2020) Reanalysis fields capturing the same sub-planetary and synoptic-scale flow features under consideration here.

Hence, we proceed pragmatically, examine only levels at and below the prevailing stratopause and adopt the ERA-Interim data set. It has a horizontal resolution at polar-extratropical latitudes of (~79km) and 60 vertical levels up to 0.1mb. For gross aspects of the flow at 10mb, recourse is also made to NCEP-NCAR reanalysis fields.


## 2.2 Diagnostic Analysis.

The objective is to establish the existence, scale, structure, evolution and dynamics of the sub-planetary transient flow features. To this end we exploit the PV perspective (Hoskins et al 1983) so that the structure is defined by the three-dimensional PV distribution, its influence upon the flow and that of its sub-features is linked to the in-situ deviation of the PV from the ambient

field, and its evolution is rendered more transparent when examined on isentropic surfaces (Davies et al, 2024).

Further insight is provided by examining, where appropriate, aspects of the contemporaneous distribution of other flow variables including the relative vorticity, divergence, potential temperature, specific humidity and ozone mass mixing ratio. For each case study we present a summary of key features at the 10mb level followed by examples of the structure and evolution of the flow at upper stratospheric levels for specific time windows.


## 2.3 Features of a Mature SPV

The Boreal Winter of 2004-2005 was comparatively non-descript and there wasn't an SSW event. At the 10mb level the time-latitude pattern of zonal mean velocity (Fig. 1a) increased almost monotonically to a maximum of ~70 ms$^{-1}$ by the 19$^{th}$ of January and decreased almost uniformly thereafter. The corresponding time-longitude pattern of the meridional velocity at

60N (Fig. 1b) shows evidence of a strong $m = 2$ pattern by the middle of December. Its amplitude and phase remained comparatively constant for the remainder of the Winter although it transiently weakened and retrogressed in early January. The onset of the wave and the forementioned transient retrogression was accompanied by a relatively strong upward EP fluxes (not shown).

At stratopause elevations, the winter's flow pattern in general showed notable, but customary, transient synoptic-scale

variations. Here we select the 11$^{th}$ January as a not atypical day, and Figure 2 shows the instantaneous patterns of the PV and the relative vorticity on a sequence of pressure levels at 00Z 11$^{th}$ January. At all levels there are planetary scale features (-zonal wavenumber $m = 1$-$3$) with evidence of finer-scale structures embedded within an annular-like structure. The upper two rows (panels a-d) indicate the presence within or on the periphery of the SPV of localized regions of PV maxima with co-located but more diffuse regions of positive relative vorticity. These features resemble those identified in DS24 and are sub-

planetary / synoptic in scale. The amplitude of their relative vorticity signifies, via Stokes's circulation theorem, a circulation of ~25 ms$^{-1}$. The contemporaneous existence of these sub-planetary scale stratospheric (SSS) features is an integral dynamical aspect of the SPV's overall structure, and their amplitude and lateral width are an indicator of the in-situ jet's sharpness. In




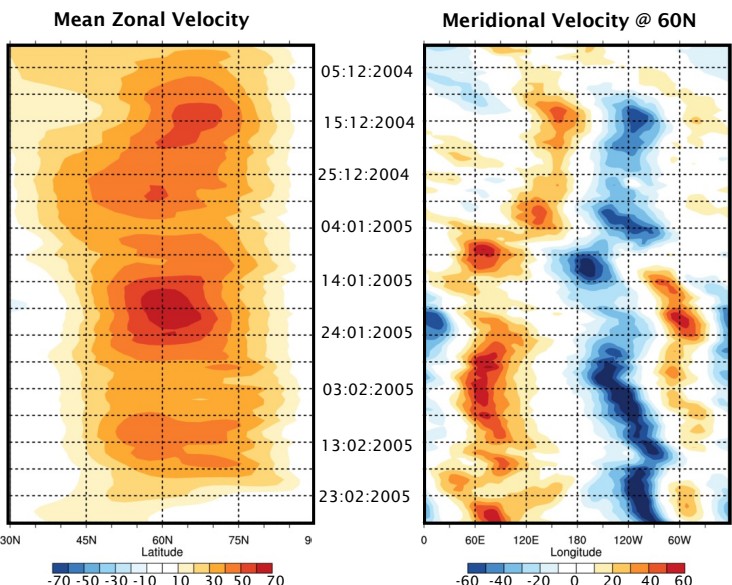

Fig. 1    Panel (a) shows the zonal mean velocity as a function of latitude (60-90N) at 10hPa for the Boreal Winter of 2004-2005, and panel (b) depicts the meridional velocity field at 60N at the same level for the same winter season. (Based upon the NCEP/NCAR Reanalysis.)





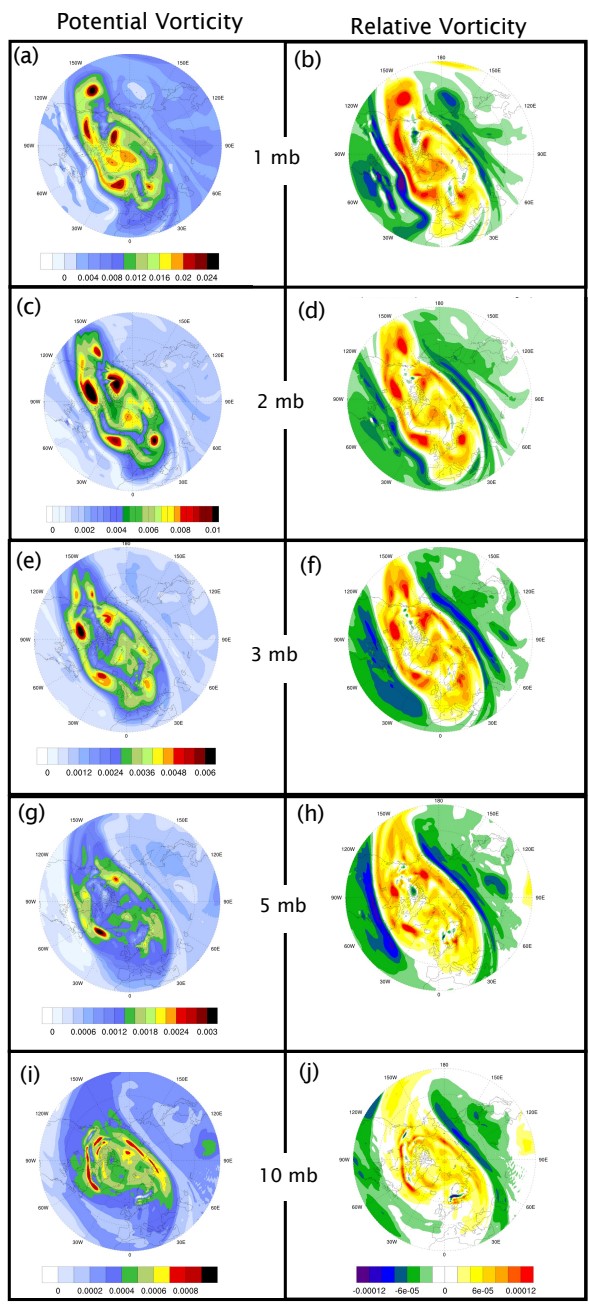






Fig. 2    Depicted are polar stereographic projections poleward from 30N of the PV in units of K m$^2$ kg$^{-1}$ s$^{-1}$ (left column) and the relative vorticity in units of $10^{-4}$ s$^{-1}$ (right column) for 00Z 11$^{th}$ January 2005 on the 1, 2, 5 and 10 mb surfaces. (Based upon the ERA-Interim Reanalysis.)


principle their presence can also influence the spatial distribution of chemical constituents within the SPV's core and at its periphery.

Panels (e-j) demonstrate that these SSS features are vertically coherent extending down to 10 hPa (~ 30 km) with the relative vorticity exhibiting comparable amplitude (~$1.0*10^{-4}$ s$^{-1}$) at all levels. Although they are deep features, their meridional
width diminishes downwards exhibiting a stalactite-like structure.

The character of the SSS features, with their predominantly positive maxima of PV, differ intrinsically from the patterns of the corresponding fields astride the tropopause level jet (Appenzeller and Davies, 1992), and this points to the different transient dynamics of the two jets.

Figure 3 displays, for the same time instant as Fig. 2, the distributions of the temperature on the 1, 2 and 3mb pressure
surfaces (panels a-c), the wind vector field on the 2mb surface (panel d), the specific humidity (panel e) and the divergence patterns (panel f). The temperature fields are comparatively smooth and indicate that the SPV's cold pool weakened above 2 mb. In general, they do not exhibit the finer-scale features of the corresponding PV and relative vorticity patterns, and this is in harmony with the vertical coherence of the latter. More trenchantly the static stability at the PV maxima (not shown), determined in terms of the potential temperature difference between the different pressure levels, is typically twice that of the
adjacent areas.

The wind vector field establishes the general location of the jet and its strength suffices to smooth the wind signature of the SSS features. In passing we note that a convenient *ad hoc* measure of the SPV's core is the domain located poleward of zero relative vorticity (- see Fig. 2 and Fig. 3d). In the ERA-Interim Reanalysis the specific humidity is inert and merely advected with the in-situ winds and hence is akin to a flow tracer. Its distribution (Fig. 3e) is suggestive of a well-mixed SPV
core that is substantially encircled by an elongated streamer of low humidity values advected on a different potential temperature surface.

The amplitude of the divergence field (Fig. 3e) is typically less than a quarter of the corresponding relative vorticity (and absolute vorticity) values, and its spatial pattern is reminiscent to that of buoyancy/gravity waves. This suggests that the dynamics of the SSS features is predominantly quasi-barotropic, albeit with a balanced flow state beyond that of quasi-
geostrophy.

The prior development of the PV pattern in Fig. (2c) is shown in Fig. (4). It displays a 6 hourly time-sequence of the corresponding patterns to 00Z 11$^{th}$ January. The fragmented annulus is seen to evolve with time and the individual embedded SSS features are readily identified exhibiting a measure of temporal coherence as they are advected by and modify the in-situ flow.




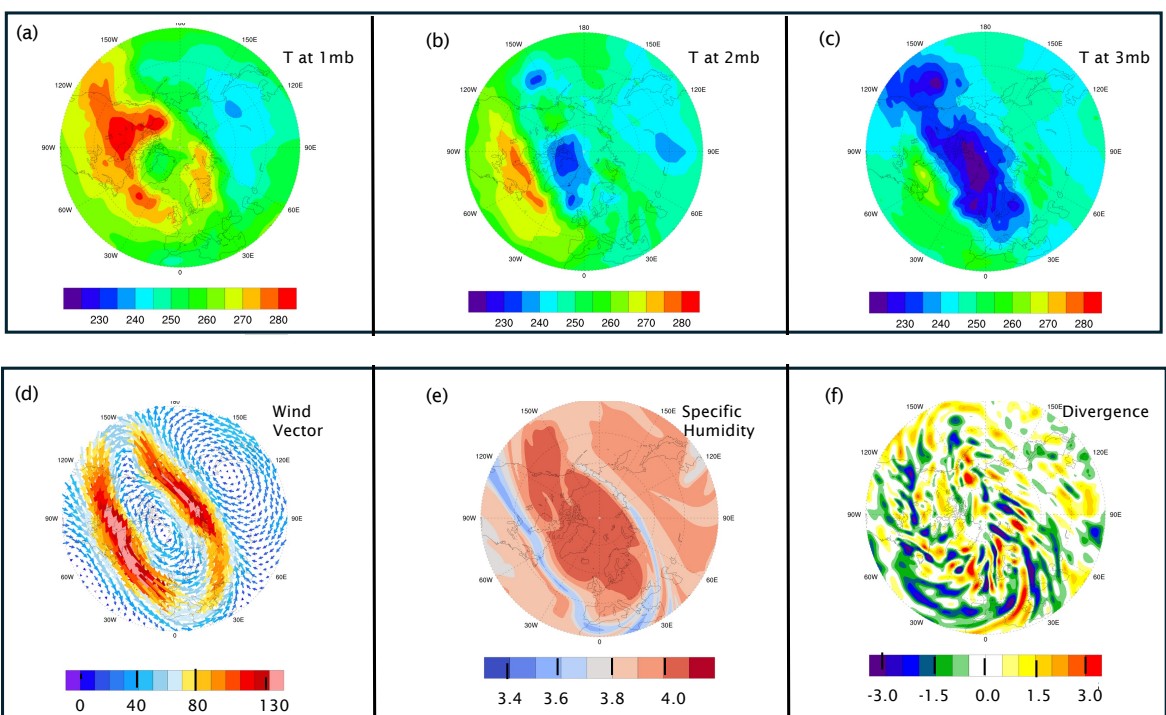


Fig. 3    Distribution of the temperature on the 1, 2 and 3mb surfaces (upper row) and the wind vector (ms$^{-1}$), specific humidity (10$^{-6}$ kg kg$^{-1}$) and divergence (10$^{-5}$ s$^{-1}$) on the 2mb surface (lower row) for the same time instant as Fig. 2. (Based upon the ERA-Interim Reanalysis.)




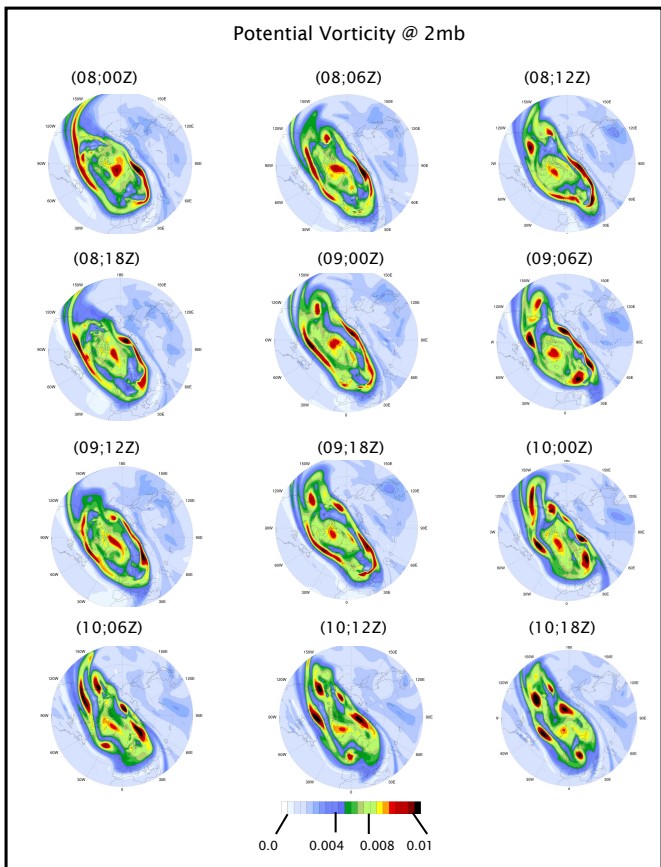


Fig. 4    Evolution of the PV (K m$^2$ kg$^{-1}$ s$^{-1}$) on the 2hPa pressure surface at 6 hourly intervals from 00Z on the 8$^{th}$ January to
         18Z on the 10$^{th}$ January.




## 2.4 Features of a nascent SPV

The Austral winter stratosphere is less subject to PRW forcing from below, and here we examine the SPV during late Spring
/ early Southern Hemisphere winter 2018. On the 10mb pressure surface, the nascent SPV in April has a weak pole-centred
PV anomaly, and by mid-May the jet's mean zonal flow had increased to ~60 ms$^{-1}$ (Fig. 5a). In late April through early May
there is evidence of a weak eastward propagating $m = 1$ wave (Fig. 5b) but it disappeared by mid-month. The accompanying
polar temperature had decreased by mid-May to ~200K with a pole-to-30S temperature difference of some 25K (not shown).

Above, at stratopause levels, the localized region of high PV at the SPV's core began to break apart in early May and
thereafter had a fragmented structure. Here the focus is on SSS features in mid-month. Fig. (6) shows the PV, relative vorticity,
divergence, potential temperature, ozone mass mixing ratio, and wind vector patterns on the 1mb pressure surface at 00Z on
19$^{th}$ May. A train of isolated PV centres circumscribes the Hemisphere near 60S with another PV feature in the neighbourhood
of the pole (Fig. 6a). The train is also readily identifiable in the relative vorticity pattern (Fig. 6b) with peak amplitudes less
than -1.2*10$^{-4}$ s$^{-1}$ - a value an order of magnitude stronger than that of the less organized divergence / gravity-wave field (Fig.
6c). The depth of the train itself is vouched by the existence (- not shown) of similar co-located patterns on the 2mb and 3mb
surfaces.

The potential temperature (θ) field (Fig. 6d) shows that at the 1mb level the train is accompanied by a co-located
zonal band of low θ (~1650K). One inference is that the forementioned polar PV feature higher θ (~ 1800K) is located on a
higher isentropic surface (say, θ~1800K). Saliently, the contemporaneous distribution of the ozone mass mixing ratio (Fig.
6e), shows not only a rich pattern of features with lower O3 values within the SPV's core, but also a string of high values
coincident with the vortex train itself. Finally, the wind vector pattern (- Fig. 6f ) is consistent with the existence of strong
shear on the poleward side of the jet and the presence of troughs linked to the stronger vortices of the train.

Figure (7) gives an indication of the prior development of the SSS features in the period from the 15$^{th}$ to the 19$^{th}$ May.
It depicts at daily interval the PV pattern on the 1700 isentropic surface and shows that the features advect eastward with and
are deformed by the in-situ flow. To a measure they retain their amplitude pointing to their adiabatic character. Their temporal
development is reminiscent of the downstream growth of a vortex train forming on a barotropically unstable shear layer rather
than a synchronous development of vortices on such a layer.






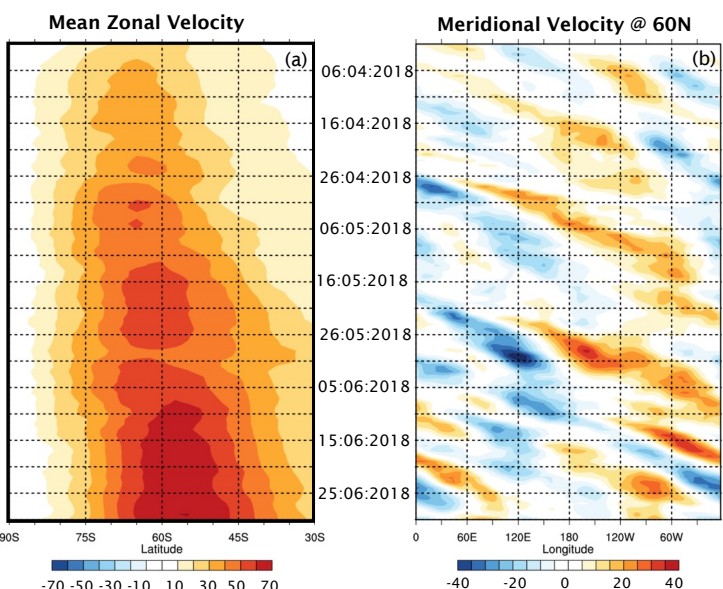

Fig. 5 Counterpart of Fig. (1) but for April-June 2018 in the Southern Hemisphere at 10mb. The zonal velocity now pertains to the latitude range 30-90S, and the meridional velocity is for 60S.





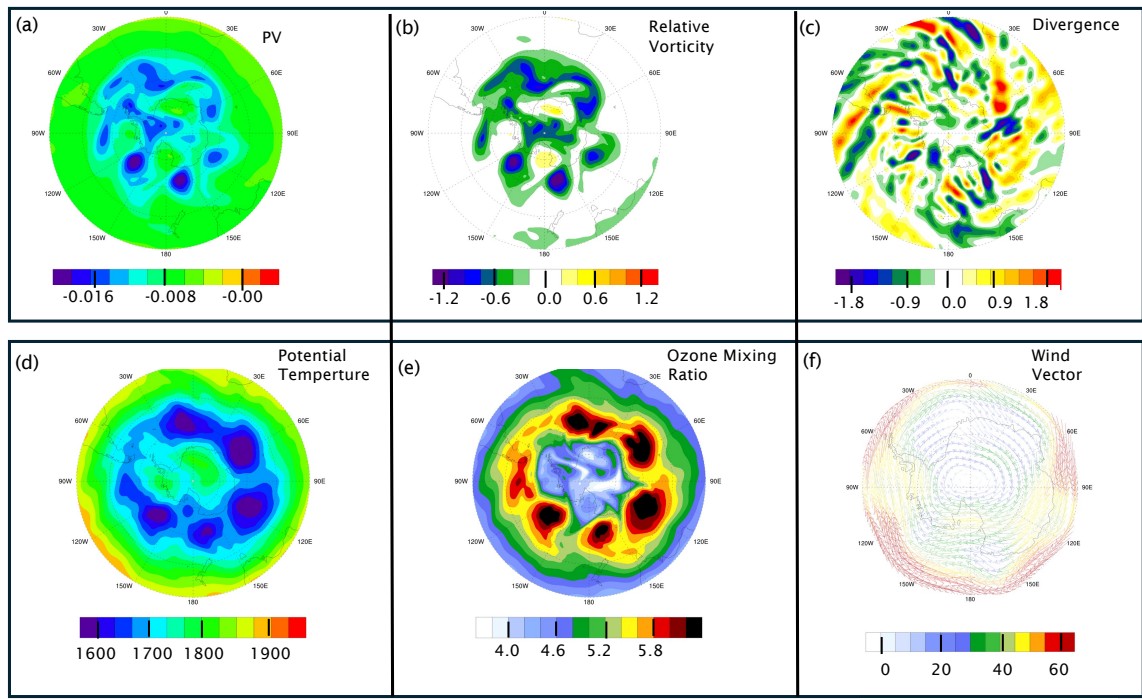


Fig. 6    Depiction of the PV(Km²kg⁻¹s⁻¹), relative vorticity (10⁻⁴ s⁻¹), divergence (10⁻⁵ s⁻¹), potential temperature (K), ozone mass mixing ratio (10⁻⁶ kg kg⁻¹), and wind vector (ms⁻¹) patterns on the 1mb pressure surface at 00Z on 19ᵗʰ May. The panels are polar stereographic projections poleward from 30S, except for panel (f) which is only poleward from 60S.





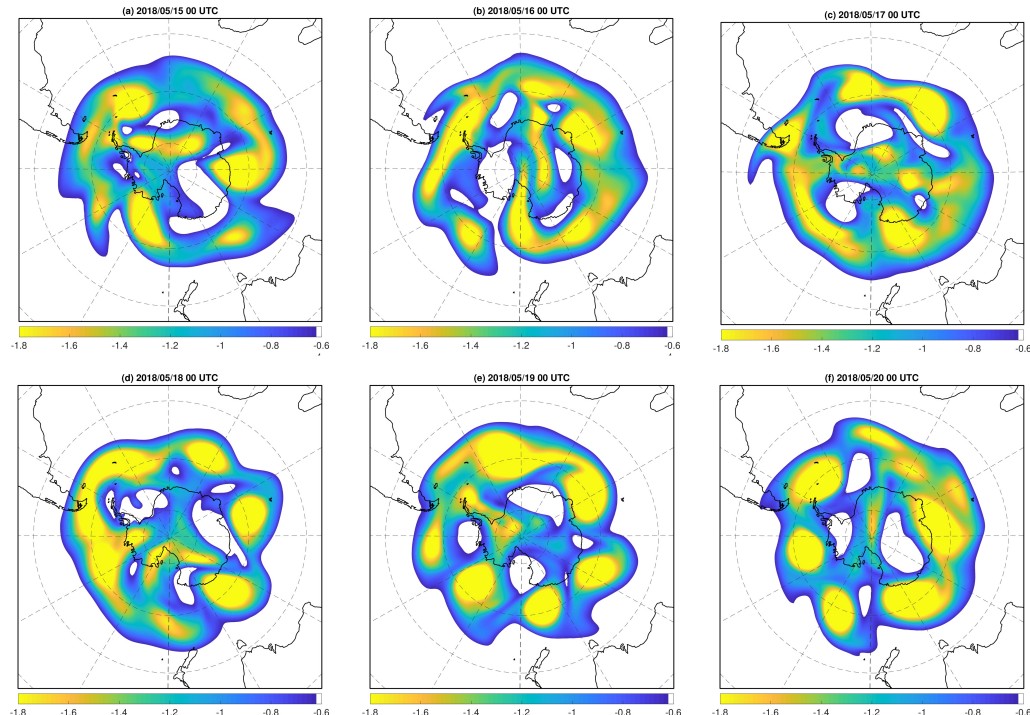


Fig. 7   Evolution of the PV pattern at stratopause levels on the 1700K isentropic surface at daily intervals from 00Z on the 15th May to the 19th May 2018. (Units: Km$^2$kg$^{-1}$s$^{-1}$ $*10^2$)





**2.5 Features related to an SSW.**

A notable Sudden Stratospheric Warming occurred in February 2018, and it has subsequently been the subject of considerable scrutiny. This includes consideration of the event's development (Butler et al 2020), the contribution of GWD (Watanabe et al, 2022), its dynamics and predictability (Rao et al 2018; Karpechko et al. 2018; Knight et al 2020; Erner et al, 2020), the

accompanying middle atmosphere chemical signature (Pérot and Orsolini, 2021), its difference to the 2019 SSW (Butler et al 2020), its impact upon the mesosphere (Wang et al, 2019), and role of surface conditions in its occurrence (Lü et al, 2020 ). Here we briefly consider the planetary-scale and annular features of this SSW.

Figure (8) is the counterpart of Fig. 1 but for the 2017-2018 Boreal Winter. It shows that at 10mb the zonal mean velocity (Fig 8a) strengthened during the last week of December, attained a maximum of ~60 ms$^{-1}$ in early January,

deaccelerated to ~30 ms$^{-1}$ in the period from the 14$^{th}$ to the 19$^{th}$ of the month, and subsequently oscillated about this value before finally reversing around the 11$^{th}$ February. The time-longitude section for the meridional velocity at 60N (Fig. 8b) indicates the presence of significant planetary scale wave patterns. The *m=1* wave present in December was supplemented by an *m = 2* signal in January and subsequently supplanted by the latter ahead of the SSW event. Polar temperature tended to decrease to the 10$^{th}$ January, increased by ~15K by the 19$^{th}$, and oscillated thereafter until the onset of the SSW (not shown).

Above, at stratopause levels, the presence of the forementioned planetary scale waves were evidenced first by an off-pole PV pattern in December, and a dumb-bell off-pole pattern in January that distorted in the run up to the SSW. Figure 9 shows the patterns of the PV (left column), relative vorticity (centre column) and temperature (right column) on the 1mb (upper row), 2mb (middle row) and 10mb (lower row) surfaces at a time (00Z 12$^{th}$ Feb.) shortly after the SPV had satisfied the standard empirical criteria for an SSW. A distinctive and strong split of the SPV's PV distribution is evident at all three levels.

The PV patterns on the upper two levels are similar (- as are the relative vorticity patterns), and the range of the temperature fields are also similar. The near-pole temperatures are relatively low (high) at the upper levels (lower level). Together these flow and thermal patterns attest to the vertical coherency of an anomalous PV structure that is 'effectively' centred at around 2mb.

Figure 10 portrays the development of the PV pattern at upper and mid-stratospheric levels at 48-hour intervals

immediately ahead of the satisfaction of the SSW criterion. The upper row depicts the patterns on the 1500K isentropic surface at 00Z on the 6, 8 and 10$^{th}$ February, and the lower row shows the corresponding patterns on the 850K isentropic surface.

On the higher surface an annular-like structure is evident on the 6$^{th}$ in the form of an off-pole elliptical structure with embedded SSS features. It begins to fragment between the 6$^{th}$ and 8$^{th}$ and by the 10$^{th}$ notable PV maxima have formed over Canada and western Europe. On the lower theta surface, the initially compact elliptic PV structure (Fig. 10d), first distends

(Fig. 10e) and then splits (Fig. 10f). Inference are that the vortex is disrupted earlier and more severely on the warmer surface.

The corresponding thermal patterns (not shown) on the upper (2mb) and lower (10mb) levels possess a dipole structure of opposite polarity that reverses with time. From the 10$^{th}$ to the 12$^{th}$ February the near polar-temperature changes at the 1, 2, and 10mb levels are respectively -18K, -32K, and +16K.




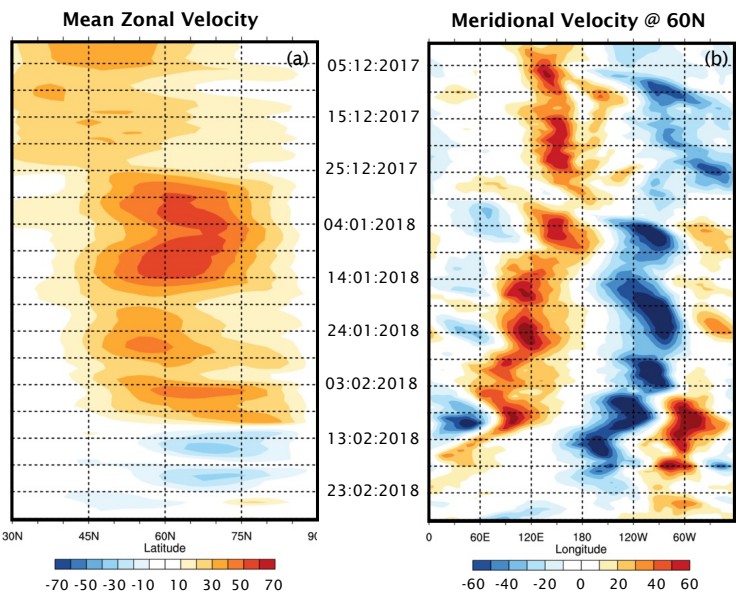


Fig. 8    As for Fig. (1) but for the Winter of 2017-2018.






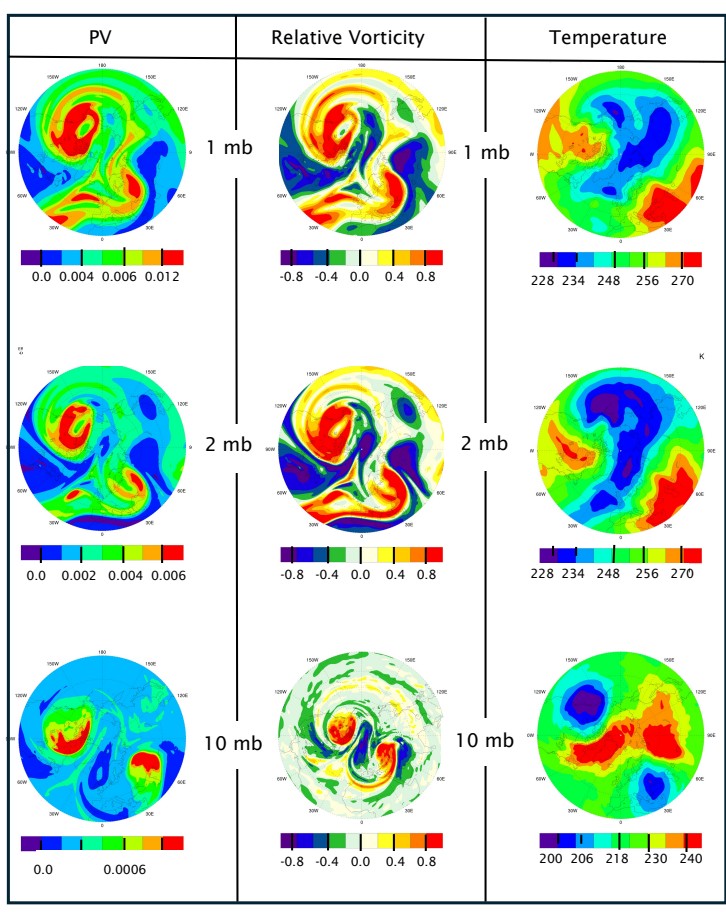

Fig. 9    Distribution of the PV(Km$^2$kg$^{-1}$s$^{-1}$) in the left column, relative vorticity ($10^{-4}$ s$^{-1}$) in the centre column, and potential temperature in the right column on the 1, 2, and 10mb pressure surfaces at 00Z 12$^{th}$ February (- shortly after the SSW).



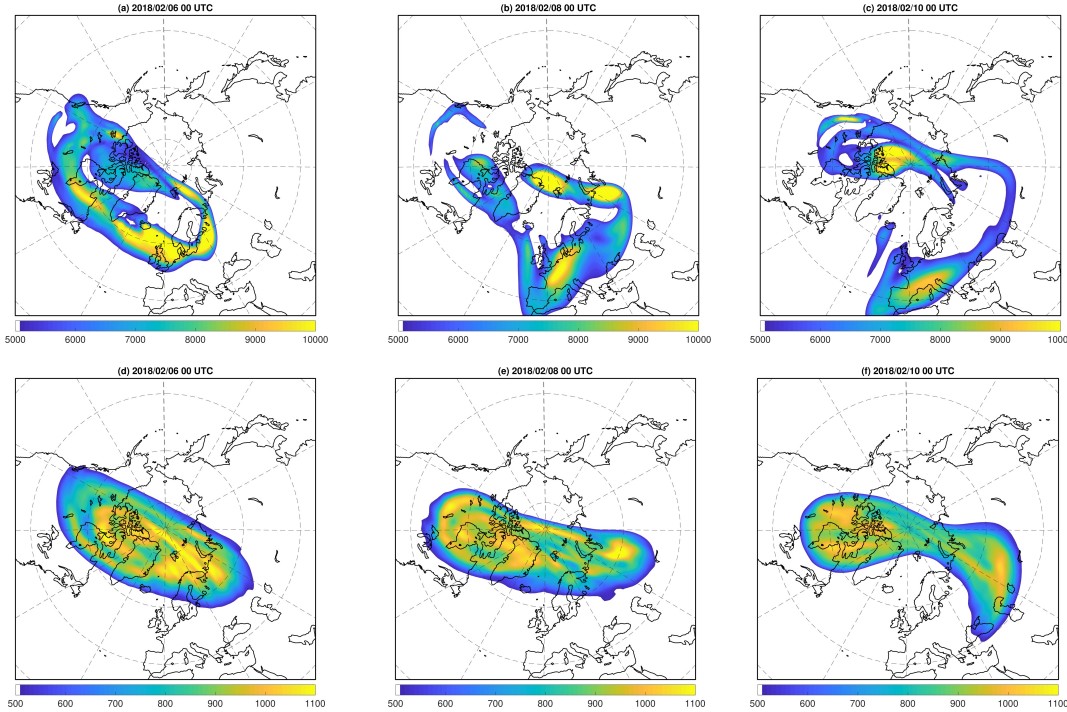

Fig. 10   The upper row (panels a-c) displays the PV patterns in the upper stratosphere on the 1500K isentropic surface at 00Z for the 6, 8 and 10$^{th}$ February, and the lower row (panels d-f) shows the corresponding pattern but on the 850K surface. (Units: Km$^2$kg$^{-1}$s$^{-1}$ $*10^6$)




## 3.0      Theoretical Model Studies.

Our objective here is to explore the character of perturbations that can evolve on an annular-like flow structure akin to that of a deep, axisymmetric and undisturbed SPV. To this end we parsimoniously adopt an idealized theoretical model capable of representing fundamental features of such a setting including the occurrence of barotropic instability. The model constitutes

non-divergent barotropic flow on a polar $\beta$-plane. It can sustain a basic state of a cylindrical (possibly unstable) jet, and it incorporates a suitable representation of the latitudinal variation of the Coriolis effect. However, it excludes consideration of three-dimensional effects and therefore can only represent flows structures that are comparatively deep. These restrictions, together with the model's simple configuration, imply that our results will constitute a potentially insightful but qualitative guide rather than providing quantitative definitive statements.


### 3.1 Model Configuration and Methods of Analysis

On a polar $\beta$-plane the earth's vorticity is given by $f = 2\Omega\,[1 - \tfrac{1}{2}\,(r^*/R)^2]$, where $r^*$ denotes radial distance on the plane and $R$ refers to the Earth's radius. Then the model's governing equation takes the form

$$D(Z^*)/Dt^* = 0, \tag{1}$$

where $D/Dt^*$ denotes the material derivative for two-dimensional flow, and $Z^*$ is the 'system-relative' absolute vorticity

$$Z^* = \zeta^* - 2\Omega\,[\tfrac{1}{2}(r^*/R)^2]\,.$$

Here $\zeta^*$ denotes the relative vorticity and is such that $\zeta^* = \nabla^2\psi^*$ with $\psi^*$ denoting the stream function. In the following the dimensional starred variables are replaced by non-dimensional unstarred variables:- $r = r^*/R$, $Z = Z^*/(2\Omega)$, $\zeta = \zeta^*/(2\Omega)$, and an azimuthal velocity $V = V^*/(2\Omega R)$.

To facilitate the derivation of analytical results we let the 'system relative' absolute vorticity of the undisturbed circumpolar vortex comprise three concentric zones (see Fig.11) of respective uniform absolute vorticity ($Z_I$, $Z_{II}$, $Z_{III}$), given by

$$Z_I = 0, \qquad\qquad\qquad\qquad\qquad \text{for } r < a, \tag{2a}$$

$$Z_{II} = C\,, \qquad\qquad\qquad\qquad\qquad \text{for } a \leq r \leq b, \tag{2b}$$

$$Z_{III} = nC\,, \qquad\qquad\qquad\qquad\qquad \text{for } r > b. \tag{2c}$$

This represents a relatively quiescent innermost core $(r<a)$, an annular region $(a \leq r \leq b)$ yielding an absolute vorticity maximum, and an outer domain $(r>b)$ of weaker vorticity (i.e. $n < 0$).

The accompanying axisymmetric basic flow $(V)$ takes the form

$$V_I = (1/8)r^3, \qquad\qquad\qquad\qquad \text{for } r < a, \tag{3a}$$

$$V_{II} = \tfrac{1}{2}Cr\,[1 - (a/r)^2] + (1/8)r^3 \qquad\qquad \text{for } a \leq r \leq b, \tag{3b}$$

$$V_{III} = \tfrac{1}{2}Cr\,[n + (b/r)^2\,(\varepsilon - n)] + (1/8)r^3 \qquad \text{for } r > b, \tag{3c}$$





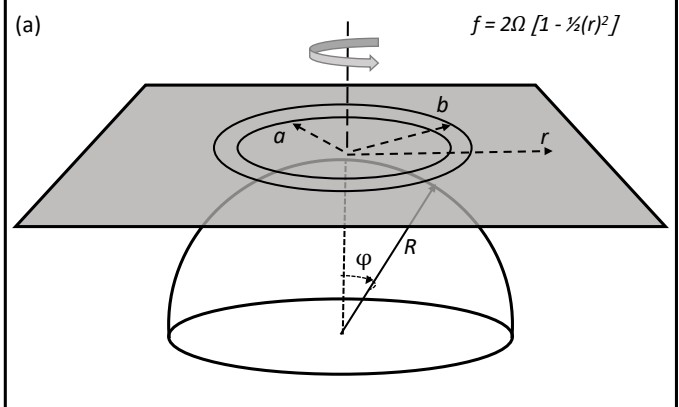

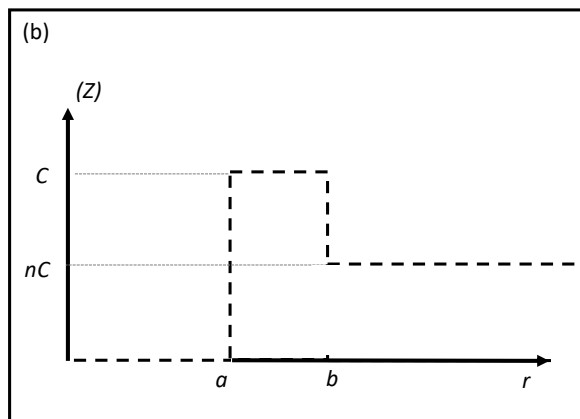

Fig. 11 Panel (a) is a schematic of the polar $\beta$-plane and the geometry for the initial distribution of the vorticity. Panel (b) shows the radial distribution of three concentric zones of uniform $Z$ with the annular band of enhanced $Z$ located between $(a < r < b)$.




It is also pertinent to note that,

$$(V_a/a) = (1/8)a^2, \qquad (4)$$

$$(V_b/b) = \tfrac{1}{2}[C\varepsilon + (1/4)b^2], \qquad (5)$$

where $(V_a/a)$ and $(V_b/b)$ denote respectively the angular velocity at $r = a$ and $b$.

Here $\varepsilon = [1 - (a/b)^2]$ is a measure of the width of the annular region relative to its radial location, and the $(1/8)r^3$ term in Eqs (3a-c) is attributable to the $\beta$-effect. In effect the parameters $(a, b, C, n)$ determine the basic state of the vortex core, with typically $C \leq 2$, $b \sim \tfrac{1}{2}$, and $a$ and $b$ such that $\varepsilon$ is capable of spanning across its permissible $[0,1]$ range.

The parameter $(nC)$ determines the relative vorticity $(\zeta)$ distribution in the exterior (i.e. $\zeta_{III} = nC + \tfrac{1}{2} r^2$). Its value is bounded by the requirements that the absolute vorticity in the far-field be less than in the polar region $(nC < 0)$, and the basic state flow should be inertially stable $(nC > -1)$. A more stringent condition is the requirements that at a radial distance, say $r = r_C$, the velocity is significantly reduced and $\zeta$ approaches zero (i.e. $nC \sim -\tfrac{1}{2} r_C^2$). For such a reduction to occur by say $\sim 30N$ then typically $0.7 > r_C > 1.1$.

Figure 12 shows illustrative examples of the basic state flow profiles $(V)$ representable by Eqs. (3). For the $r < b$ domain, the jet profiles correspond to maxima of 90 ms$^{-1}$ and 135 ms$^{-1}$ located at $b = \tfrac{1}{2}$, ($\approx 60N$) for wide ($\varepsilon = 0.91$), intermediate ($\varepsilon = 0.75$) and narrow ($\varepsilon = 0.36$) annuli. The profiles in the outer domain $(r > b)$ span those for realistic values of $r_c$. In essence the parameter settings permit a range of suitable velocity profiles.

Results are derived for the linear and non-linear response of perturbations of stipulated basic states. The linear component includes a normal mode analysis that establishes the growth rate and azimuthal wavenumber $(m)$ of unstable modes, and an analysis of the response to a specific form of non-normal mode initial perturbation. The non-linear response is simulated using a contour-dynamics technique (Dritschel, 1989) with adjustments of the standard contour advection technique to (a) account for the specification of the absolute as opposed to the relative vorticity and (b) circumvent the standard contour advection stipulation of non-zero absolute vorticity in the outermost domain. The forward integration is conducted using a 4$^{th}$ order Runge-Kutta scheme and, depending upon the purpose of the simulation, the initial state is perturbed by adding either weak amplitude random noise or a finite amplitude displacement to the location of the domain interface at $r = b$.




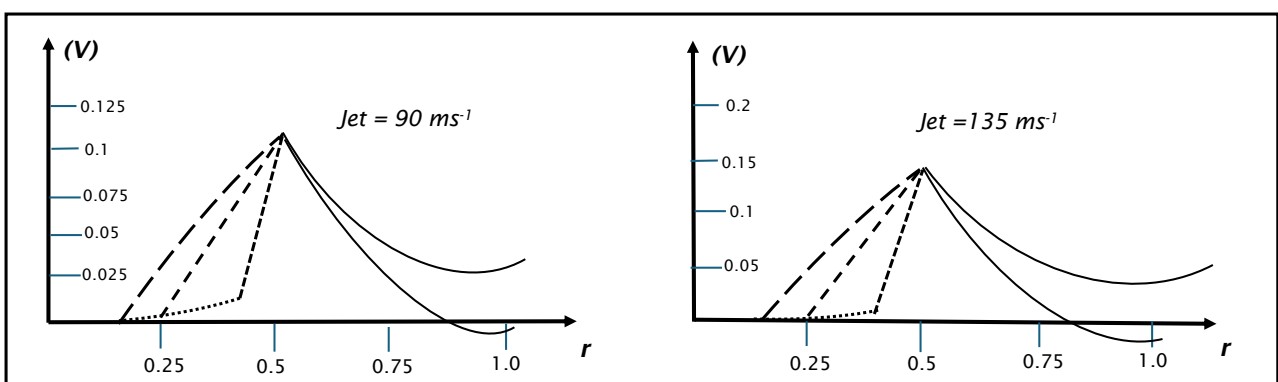


Fig. 12   A sample of the model's possible basic state velocity (*V*) profiles corresponding to a dimensional jet strength of 90
ms$^{-1}$ (left) and 135 ms$^{-1}$ (right) located at located at *b*= ½ (≈ 60N). The three curves for *r* < *b* in each panel depicted
with long, intermediate and short dashes refer respectively to wide *(ε = 0.91)*, intermediate *(ε = 0.75)*, and narrow *(ε
= 0.36)* annuli, and the dotted line is the velocity profile in the *r* < *a* sector. The two full curves beyond *r* = *b* illustrate
the physically representable band of velocity profiles.






### 3.2 Normal Mode Analysis


3.2.1    Derivation of the Instability Criterion

Small amplitude perturbations of the governing equation (Eq. 1) satisfy the linear equation,

$$\{\partial/\partial t \ + (V/r)\partial/\partial \theta \}q' + u' \ \partial/\partial r \ (Z) \ = \ 0 \qquad (6)$$

Here $(q', u')$ refer to the perturbation vorticity and radial velocity with $q' = \nabla^2 \psi'$. Likewise, $(Z, V)$ are, as indicated earlier,

the basic state absolute vorticity and azimuthal velocity (Eqs. 2 and 3).

Adopting a 'PV Perspective' for the perturbations (Davies and Bishop, 1994) solutions of Eq. (6) are sought for stream-function perturbations $(\psi')$ of the form

$$\psi' = \ \psi'_\alpha + \psi'_b,$$

where the two components represent wave perturbations propagating respectively on the vorticity discontinuities at the domain

interfaces located at $r = a$ and $b$. The associated stream functions are

$$\psi'_{(a,b)} = F_{(a,b)}(\text{r}) \ sin \ (m\theta - \omega t) \ .$$

For basic states with the form of Eqs. (3), $F$ has a $r^m$ and $r^{-m}$ dependency respectively within and exterior to the pertinent interface. On applying appropriate conditions at $r = 0$, as $r \rightarrow \infty$, and at the interfaces, the wave amplitudes at the two interfaces, $F_{(a,b)} = (A, B),$ are linked by the relationships,


$$(\omega/m)A \ = \ (V_a/a)A + \tfrac{1}{2}(C/m) \ [A + (1- \varepsilon)^{m/2}B], \qquad (7a)$$

and

$$(\omega/m)B \ = \ (V_b/b)B - \tfrac{1}{2}(C/m)(1 - n) \ [B + (1- \varepsilon)^{m/2}A] \qquad (7b).$$

The three-terms on the right hand side in each of Eqs. (7a, b) correspond to:- translation with the ambient angular velocity (Eqs. 3d, e); a 'Rossby-wave type' contribution of the in-situ wave; and a far-field 'Rossby-wave type' contribution

of the influence of a wave at one interface upon that at the other interface. In the absence of the latter far-field contributions, equivalence of the phase velocities would require - see Eqs. (4, 5)

$$\mathcal{D} = 0 \ , \qquad (8a)$$

$$\text{where } \mathcal{D} = [m \ \varepsilon \ (1+\mu) - (2- n)] \qquad (8b)$$

$$\text{with } \mu = b^2/(4C).$$

We return to consider the import of Eq. (8a) later, but proceed here to note that together Eqs (7a, b) yield the linear system's wave-dispersion relationship. After some manipulation the instability criterion can be written as

$$\Delta > 0, \qquad (9a)$$

$$\text{with } \ \Delta = \ [ \ -\mathcal{D}^2 + 4 \ (1- n) \ (1- \varepsilon)^m], \qquad (9b)$$

and for unstable waves, the growth rate $(\sigma)$ and angular phase speed $(v)$ are given by,






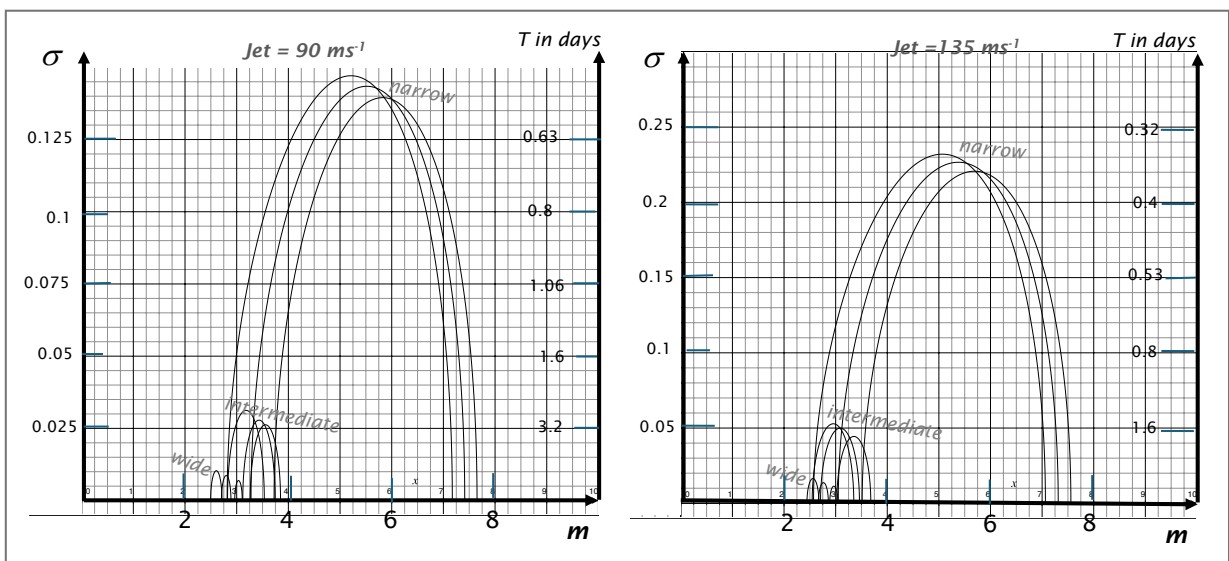

Fig.13    Growth rate ($\sigma$) displayed as a function of the wavenumber *(m)* for a jet of strength 90 ms⁻¹ (left) and 135 ms⁻¹ (right)
located at *b = ½ (≈ 60N)* The three family of curves are for the annular bands listed in Fig. (12).

$$\sigma = (1/4)\ C\ \sqrt{(\Delta)}\ , \tag{9c}$$
$$v = ½\ [(V_a/a + V_b/b) + (nC)/m], \tag{9d}$$

In line with the necessary criteria for barotropic instability the annular region needs to be a maximum of absolute vorticity (see
Eq. 9b), i.e. *(1- n) >0.*



### 3.2.2    Sample Growth Rate Curves.

The general dependency of the instability upon the configuration of the basic state is illustrated in Fig. 13. It shows the growth rate ($\sigma$) as a function of the wavenumber *(m)* for the settings corresponding to Fig.12. The growth rate increases markedly with jet strength, decreases strongly with the width of the annulus, and is comparatively insensitive to the radial velocity decay of in the outer region. In effect the sensitivity to jet skewness is primarily governed by the structure within the core. These results

are in qualitative agreement with, and complement those of, earlier barotropic instability studies (see in particular Hartman, 1983) but with some differences for the graver modes (- see later). The large growth rate of the sub-planetary modes is notable since a pragmatic lower bound for an unstable mode's growth rate would be that $\sigma$ should exceed $\tau$, where $\tau \sim (10\ \text{days})^{-1}$ is the radiative decay time-scale (Shine, 1987) of deep perturbations within the polar-night vortex in the upper stratosphere.

### 3.2.3    An Instability Ready Reckoner and its Interpretation,

Figure 13 provides a useful, but non-exhaustive, display of instability's sensitivity to the basic state configuration. To pinpoint the dependencies illustrated by, and dynamics underlying, the results displayed in Fig. 13, we resort to the PV-based phenomenological interpretation (Lighthill, 1967; Hoskins et al 1985) and mathematical formulation (Davies & Bishop, 1994) of quasi-geostrophic instability. In this framework the present instability is viewed, in terms of the two waves propagating

respectively on the opposite-signed PV jumps at *r=a* and *r=b* , and held stationary relative to one another by the differing in-situ flow at their in-situ locations. In such a configuration the waves inter-dependency can promote mutual amplification and modification of their azimuthal phase velocities.

Noting that $\mathcal{D} = 0$ implies that phase locking of the two waves is achieved without an inter-wave contribution so that the two perturbations are in quadrature and their influence is constrained to yield maximal growth. From Eq. (7) we see that a

mode with a small (large) wavenumber yields a large (small) in-situ Rossby-type contribution to the phase speed rendering it more difficult (easier) to phase-lock the two interface perturbations.

This phase locking mechanism then forms the basis for an instability 'Ready Reckoner'. The condition $\mathcal{D} = 0$ implies (Eq. 9b) that there is an azimuthal wave number ($m_0$) for which a wave perturbation is unstable with

$$m_0 = (2 - n) / [\varepsilon (1+\mu)] \tag{10a}$$

In effect $m_0$ is the ratio of the sum of the two vorticity discontinuities to the difference in the angular velocities across the annular region. It can be reformulated as

$$m_0\,\varepsilon = 2\ [1 + r_c^2/(4C)]/[1 + b^2/(4C)] \qquad . \tag{10b}$$

Thus, for specified values of the external parameters, satisfaction of Eqs (10b) signifies the instability of the wavenumber $m_0$. The accompanying growth rate is given by Eq. (9c), so that

$$\sigma = \tfrac{1}{2}\ C\ [(1 + \tfrac{1}{2}(r_c^2/C)\ (1 - \varepsilon)^m]^{1/2} \tag{11}$$


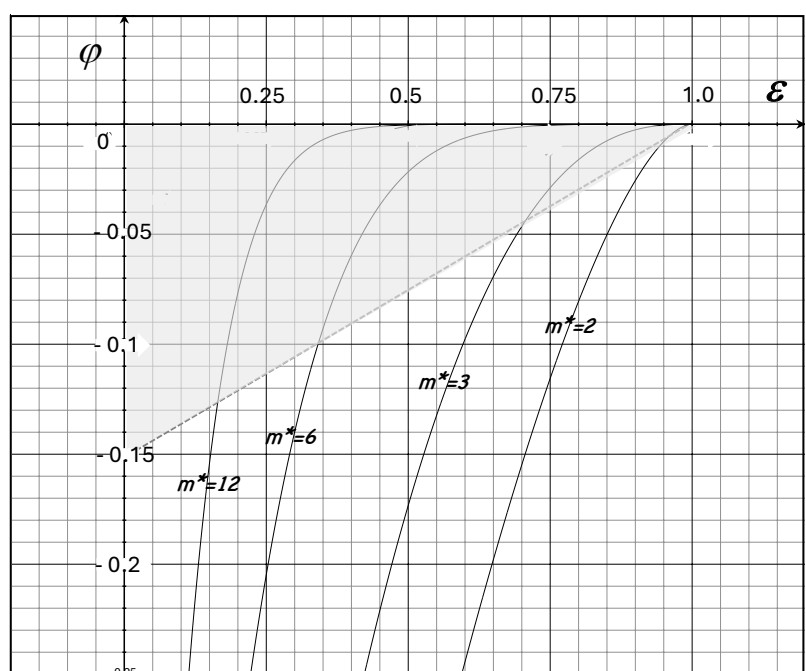

Fig. 14 The function $\varphi = \varphi(\varepsilon)$ displayed for selected values of the wave number $m^*$. The permissible domain is confined to the lightly shaded wedge region defined by $(m\,\varepsilon) > 2.0$.


More strikingly, $m_0$ also provides (for $m_0 > 2$) an excellent estimate of the wavenumber $(m = m^*)$ of the most unstable mode for
the same configuration because





$$(m^*/m_0) = 1 + \lambda \, \varphi \tag{12}$$

$$\text{where} \quad \lambda = [1 + \tfrac{1}{2} r_c^2]/\{[1 + r_c^2/(4C)]\,[1 + b^2/(4C)]\}$$

$$\text{and} \qquad \varphi = (1 - \varepsilon)^{m^*} \ln(1 - \varepsilon)/(\varepsilon).$$

with $\lambda \approx 1$ for the range of pertinent flow configurations and $\varphi \sim - 0.1$ (see Fig. 14). Thus, the wavenumber of the most unstable
wave is slightly less that indicated by Eq. 10 (i.e. $m^* < m_0$), but the difference is typically of the order of, or less than, 15%.

Thus Eqs. (10, 11) constitute a 'Ready Reckoner' for the most unstable mode and its growth rate. The sensitivity of
this seminal mode to the externally specified configuration is displayed in Fig. 15. It displays $(m_0\varepsilon)$ as a function of $C$ for the
indicated values of $(r_c, b)$. For $C \gg 1$ the most unstable wavenumber is given by $m^* \approx 2/\varepsilon$. Contrariwise for $C \to 0$ then $m^* \to$
$2(r_c/b)^2$, and the accompanying growth rate $\sigma \approx [(1/8)(r_c^2 C)(1 - \varepsilon)^m]^{1/2}$ will be comparatively weak.

These deductions indicate, in conformity with Figs. 13 and 15 that :- there is a preferred mode that maximizes growth;
the occurrence of unstable large wavenumbers ($m^* > 8$) is favoured by narrow annular regions (- small $\varepsilon$) and strong jets;
unstable sub-planetary scale waves $(3 < m^* < 8)$ exist for a wide range of external settings; and unstable planetary-scale
wavenumbers $m^* = 1-3$ only exist, if at all, for wide annular regions (large $\varepsilon$) and broad jets. The internal ratio $b^2/(4C)$ has a
dichotomous role supporting the possible occurrence of the small wavenumber modes but concomitantly reducing their growth
rate.

In effect the Ready Reckoner is consistent with, and provides a compact account of, the results displayed in Fig. 13.
A further deduction is that, in conformity with the cases study examples of Section 2, sub-planetary and synoptic-scale features
can evolve preferentially for a wide-range of realistic annular flow settings.

### 3.2.4    Stability of the m = 1, 2 Modes

The gravest modes ($m = 1, 2$) are of particular interest in connection with the occurrence of an SSW and therefore merit specific
attention. For these two modes the instability criterion, *sic.* the parameter $\Delta$ (Eqs. 9), equates to,

$$\Delta = - \{ \varepsilon^2(1 + \mu)^2 - 2\varepsilon[n + \mu(2-n)] + n^2\} \qquad \text{for } m=1,$$

$$\text{and} \qquad \Delta = - \{ \varepsilon^2[\mu(2+\mu)+n] - \varepsilon[n + \mu(2-n)] + (n^2/4)\} \qquad \text{for } m=2,$$

where, as before, $\mu = b^2/(4C)$ and $n = -\tfrac{1}{2}(r_c^2/C)$. Inspection shows that generally $\Delta < 0$ for realistic values of $\mu$ and $n$, so that
both modes are indeed stable. Notwithstanding $\Delta$ can be small ($0 > \Delta > - 0.2$) for $m = 2$ across the entire range of $\varepsilon$ values, and
small for $m = 1$ as $\varepsilon \to 0$. In the next sub-section, we consider a mechanism that runs counter the non-growth of these normal
modes.






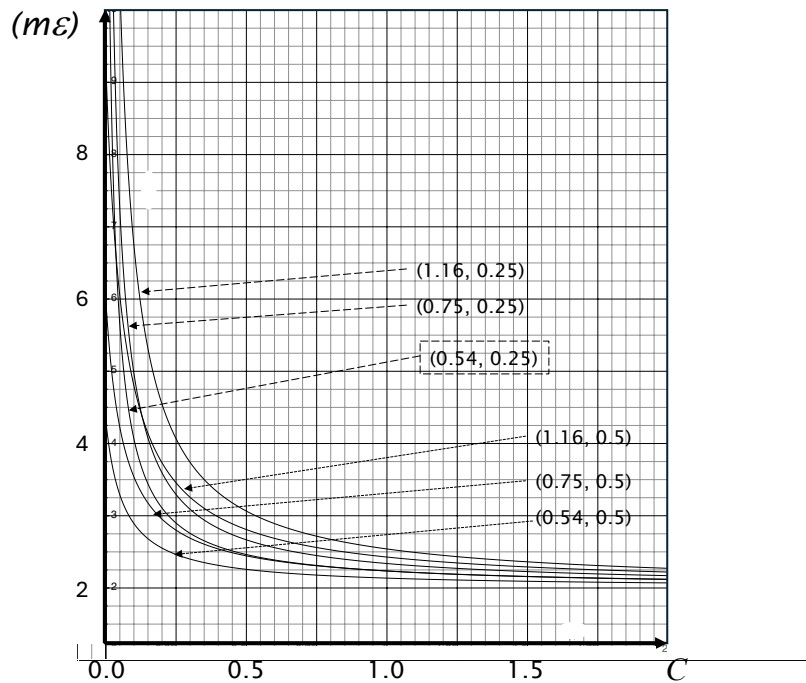

Fig. 15 A sensitivity guide to the most unstable wave to the specification of externally specified parameters. Displayed is a plot of $(m_0\varepsilon)$ as a function of $C$ for the indicated values of $(r_c., b)$.

### 3.3 Initial-value Dependency.

Consider a configuration of our model system that at the initial instant comprises a pre-existing small amplitude wave on the

outer $(r = b)$ interface but with no perturbation on the inner $(r = a)$ interface. The analogue stratospheric setting would be the





combination of:- (a) the distortion of the outer interface by, say, a Rossby wave emanating from tropopause elevations and propagating upward preferentially along the waveguide formed by the sharp PV gradient on the polar vortex's outer edge; and (b) the inner interface being initially substantially undisturbed being protected by the reversed latitudinal PV gradient on an annular band's inner side.

To replicate the model's evolution of such a configuration we now allow the stream functions of the two interface waves to assume the form,

$$\psi'_1 = \mathcal{A}(t)\ sin\ [m\theta - \delta_1(t)]\ , \qquad\qquad on\ r=a \qquad\qquad (13a)$$

and        $$\psi'_2 = \mathcal{B}(t)\ sin\ [m\theta - \delta_2(t)]. \qquad\qquad on\ r= b \qquad\qquad (13b)$$

In effect the waves are allowed to have a time-varying amplitudes ($\mathcal{A}$, $\mathcal{B}$) and phases ($\delta_1$, $\delta_2$) at respectively $r = (a, b)$, with

$\mathcal{B}(0) = \mathcal{B}_0$ and $\mathcal{A}(0) = 0$.

To replicate the model's evolution of such a configuration we now allow the stream functions of the two interface.

Substitution into Eq. (6) yields the following triad of predictive equations,

$$\partial/\partial t\ (\mathcal{A})\ =\ -\ \tfrac{1}{2}\ (C\gamma)\ \mathcal{B}\ sin\ \delta\,, \qquad\qquad (14a)$$

$$\partial/\partial t\ (\mathcal{B})\ =\ -\ \tfrac{1}{2}\ (C\gamma)\ (1-n)\ \mathcal{A}\ sin\ \delta\ , \qquad\qquad (14b)$$

$$\mathcal{A}\mathcal{B}\ \partial/\partial t(\delta)\ =\ -\ \tfrac{1}{2}C\ \{(\mathcal{A}\mathcal{B})\ \mathcal{D}\ -\ \gamma\ [(1-n)\mathcal{A}^2 + \mathcal{B}^2]\ cos\ \delta\}, \qquad\qquad (14c)$$

where $\gamma = (1 - \varepsilon)^{m/2} = (a/b)^m$, and $\delta$ is the relative phase-difference $\delta = (\delta_1 - \delta_2)$.

Equations (14a, b) imply that the inner and outer waves can grow synchronously provided that (i) $(1- n) > 0$ irrespective of the value of $\Delta$, and (ii) $\delta$ lies in the range $[-\pi, 0]$ with maximum growth prevailing when $\delta = -\pi/2$.

The triad possess two temporal invariants,

$$\partial/\partial t\ [(1-n)\mathcal{A}^2 - \mathcal{B}^2] = 0, \qquad\qquad (15a)$$

and        $$\partial/\partial t\ \{\tfrac{1}{2}(\mathcal{D}/\gamma)\ [(\mathcal{A}^2 + \mathcal{B}^2)/(2-n)]\ -\ \mathcal{A}\mathcal{B}\ cos\ \delta\} = 0. \qquad\qquad (15b)$$

These invariants can be exploited to show that the growth of the wave on the inner interface $(r = a)$ is given by,

$$\partial/\partial t\ (\mathcal{A})\ =\ (1/4)C\ [\Delta\mathcal{A}^2 + 4\gamma^2\mathcal{B}_0^2]^{1/2} \qquad\qquad (16)$$

so that        $$\mathcal{A}\ =\ (2\gamma/\Delta^{1/2})\ \mathcal{B}_0\ sinh\ [\Delta^{1/2}\ T] \qquad for\quad \Delta > 0 \qquad\qquad (17a)$$

and        $$\mathcal{A}\ =\ (2\gamma/\Delta^{*1/2})\ \mathcal{B}_0\ sin\ [\Delta^{*1/2}\ T] \qquad for\quad \Delta^* = -\Delta > 0 \qquad\qquad (17b)$$

$$where\ T = (1/4)Ct$$

Note that for an unstable configuration $(\Delta > 0)$, the factor $\Delta^{1/2}T$ is the normal mode growth rate (Eq. 9c), and the coupled waves evolve toward a state of exponential growth. For a stable configuration $(\Delta^* > 0)$ the amplitude attains a maximum of

$$\mathcal{A}_{max} = 2(a/b)^m\ \mathcal{B}_0\ /\Delta^{*1/2} \quad at\ the\ time\ t\ =\ 2\pi/(C\Delta^{*1/2}\ )\ . \qquad\qquad (17c)$$




605 For both settings, the initial short time evolution is given by

$$\mathcal{A} \approx 2(a/b)^m \mathcal{B}_0 \, T \qquad , \qquad\qquad\qquad (17d)$$

The seminal deduction is that the presence of an annular band leads to an initial secular growth of the $\mathcal{A}$-wave and that trenchantly, even for a stable basic state configuration, it is independent of $\Delta^*$. Moreover, the dependency upon $(a/b)^m$ indicates that it is larger for a narrow annular region and planetary-scale modes. Pointedly this secular growth can be

610 significant. For example, if $(a/b) = \frac{1}{2}$ and $C = \frac{1}{2}$, then in dimensional terms $\mathcal{A}$ would attain the value of $\mathcal{B}_0$ in $\sim 2/3$ day for $m = 1$ and $\sim 4/3$ days for $m = 2$, so that the wave-mean flow interaction would rapidly become non-linear. For the stable configuration, a large value of $\mathcal{A}_{max}$ requires in addition that $\Delta^*$ be small. Together Eqs. (17c, d) establish the form of the basic state that would yield significant transient growth of the $\mathcal{A}$ wave.

The accompanying amplitude of the $\mathcal{B}$-wave and the phase-difference are,

615     $\mathcal{B}^2 = (1-n)\mathcal{A}^2 + \mathcal{B}_0^2$             (17e)

and      $\cos^2\delta = (1/4)\,(\mathcal{D}/\gamma)^2(\mathcal{A}/\mathcal{B}_0)^2/[1 + (1-n)(\mathcal{A}/\mathcal{B}_0)^2]$      (17f)

Eq. (17e) implies that after significant growth the amplitude ratio of the two waves would be

$$(\mathcal{A}/\mathcal{B}) = 1/(1-n)^{1/2} = 1/[1 + r_c^2/(2C)]^{1/2}$$

so that for a weak vortex the $\mathcal{B}$ wave would dominate. A feature of Eq. (17f) is that, for a flow setting with $\mathcal{D} = 0$, the $\mathcal{A}$ and

620 $\mathcal{B}$ waves are *ab initio* phase-locked with $\delta = -\pi/2$ and hence the optimum growth criterion is sustained (Eqs. 14a, b).

An implication of this analysis follows from noting that the stratosphere's transmissivity to upward propagating PRWs serves as a 'scale-selection' mechanism only allowing these gravest waves to reach well into the stratosphere and disturb the vortex's rim at higher elevations. Thus, it is the gravest waves that could initiate the fore-mentioned significant 'scale-sensitive' secular growth, despite the stability of the basic flow to low wavenumber normal-mode perturbations.


### 3.4  Non-linear Development.

For the same model configuration, we examine the non-linear evolution (Figs. 16,17) of perturbations to the model's stipulated basic state. In Figure (16) the columns depict respectively the initial small amplitude randomly perturbed state (left column), an early phase of development (central column) and the mature non-linear (right column) patterns for three different initial

630 flow settings each with a jet strength of 120ms$^{-1}$. In close accord with the linear instability theory, the upper two rows capture the development of the synoptic-scale most unstable wavenumbers ($m = 6$ and $9$) that then leads to an aggregation of the bands into quasi-isolated mono-signed vortices each attached to its neighbours by thin filaments. The third row is for a wide vortex, for which the $m = 3$ and $4$ waves (and only these wavenumbers) possess a weak instability, and after an extended time there is a complex break-up of the annular structure.




Figure (17) shows one example of the evolution of an initial finite amplitude $m = 2$ perturbations of the $r = b$ interface but with no perturbation of the inner ($r = a$) interface. This setting is that envisaged in the previous sub-section, and pertains for ($C, a, b, n$ ) = (1.0, 0.447, 0.5, -0.5) that corresponds to a jet of 120 ms$^{-1}$ is perturbed by a wave with amplitude of 0.03. Again, in harmony with the results derived earlier the evolution exhibits a notable growth over a period of 3.2 days that results


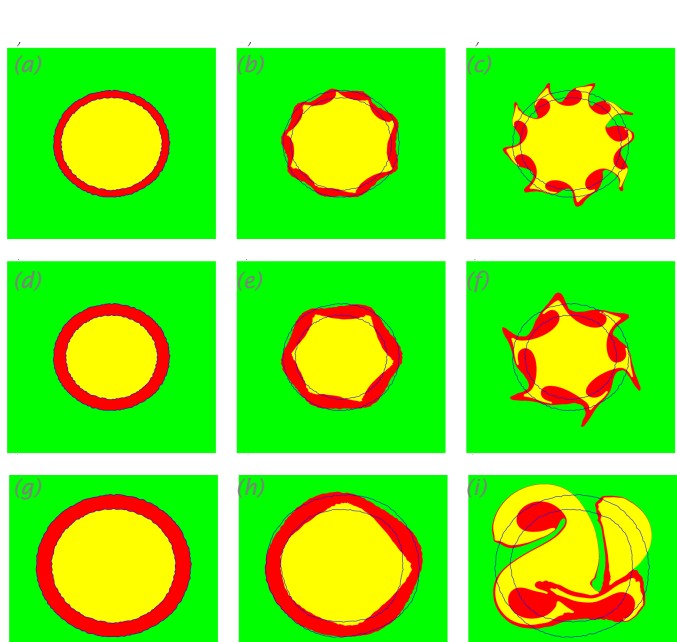

Fig. 16 Evolution of three different annular bands subjected to initial small amplitude random perturbations. The initial states are shown in panels (a, d and g) for which the external parameters *(a, b, C, n)* take the respective values of (0.434, 0.5, 1.71, -0.57), (0.4, 0.5, 1.12, -0.63), and (0.53, 0.67, 0.20, -0.65). In each case the jet strength is 120ms$^{-1}$. Panels (b, e and h) display an early quasi-linear development phase, and panels (c, f, i) capture a mature non-linear phase.






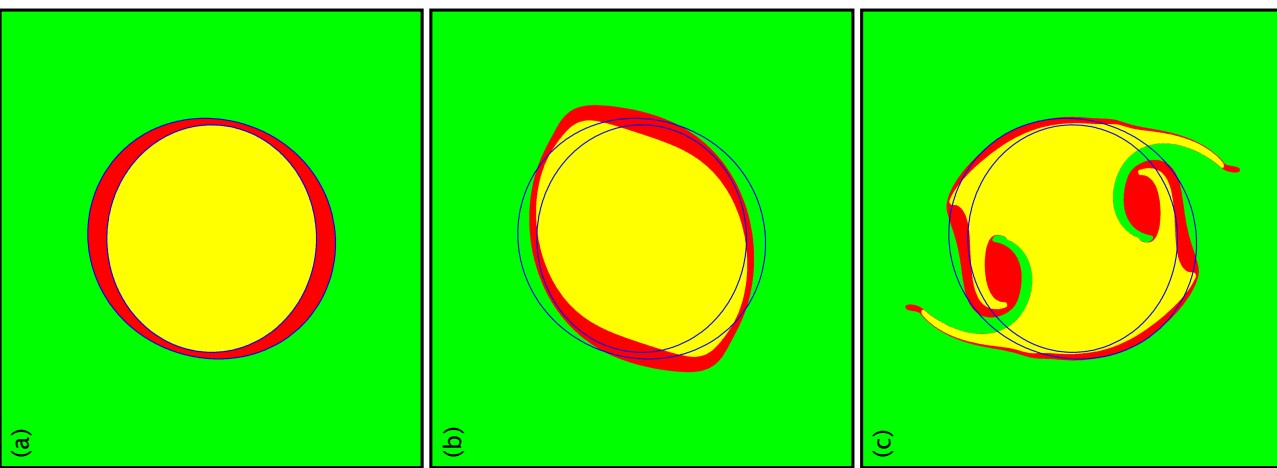


Fig. 17 Development of an annular band's absolute vorticity when the outer interface ($r = b$) is subject to an initial finite-amplitude $m = 2$ wave perturbation. The simulation is for *(a, b, C, n)* taking the values *(0.447, 0.5, 1.0, -0.5)* with an initial wave amplitude of *(0.03)*. Panel (a) displays an initial state, and the patterns displayed in panel (b) and panel (c) are in dimensional terms after 1.3 and 3.2 days.




in the annulus forming two vortex structures with each possessing a trailing filamentary structure. The pattern bears some resemblance to the split classifications that can emerge during an SSW event.

It is pertinent to note the results of two null simulations. First for an initial small amplitude random noise perturbation of the interfaces, there evolves the (expected) growth of a wave number $m = 12$. The second corresponds to an initial configuration comprising a jet of the same strength at $r = b$ but now with a uniform absolute vorticity distribution across the entire $r < b$ domain. When subjected to the same 0.03 wave perturbation at $r = b$, the configuration evolves to a steadily rotating ellipse with an aspect ratio close to unity. In effect, these 'null' simulations indicate that the vortex breakdown is

attributable explicitly to the existence of the annular band.

### 4.0       Further Remarks.

The observational component of this study is limited to the three case studies and hence can merely hint at the climatology of the detected sub-planetary features. It is constrained by the adequacy of the raw data used for the Reanalysis, and by the fidelity

of the upper levels phenomenologically based sub-grid scale Gravity Wave Drag (GWD) parameterization scheme and ad hoc divergence damping scheme of the assimilation model. In so-far as these schemes are tuned to match the expected state of the upper atmosphere, they have an artefact element that might/could compensate for or disguise other physical effects. These limitations are offset somewhat by the similarity of the features derived with differing Reanalysis models (DS24).

          The theoretical component of the study is constrained by the adoption of a non-divergent barotropic model. Positively

the model does not need to invoke quasi-geostrophy (c.f. Davies, 1981; Hartmann, 1983), incorporates an acceptable representation of the $\beta$-effect (c.f. Dritschel, 1989; Dritschel and Polvani, 1992) as opposed to spherical geometry (c.f. Hartman,1983; Michaelangeli et al.,1987; Manney et al., 1988; Mitchell et al., 2021; Waugh et al., 2023), and eminently enables the derivation and interpretation of analytical solutions for a wide range of relatively realistic basic states velocity fields. Negatively it excludes consideration of baroclinic (c.f. Dickinson, 1983; Simmons 1974) and mixed barotropic-

baroclinic (c.f. Manney & Randel, 1993) instability and is best at representing barotropic states and deep perturbations (Paldor, et al 2021). The character of the observed features signals that these shortcomings are not pathological.

          Here the normal mode graver modes ($m = 1, 2$) are stable, and contrast with the instability of some earlier studies undertaken with a special basic state structure (Hartmann, 1983). This discrepancy is attributable to a second, and generally unrealized, mid-latitude PV annulus in the special formulation.

Notwithstanding the foregoing remarks the present study lends credence to two assertions regarding the SPV. The first is that *the vortex's periphery is very frequently populated with distinctive sub-planetary scale flow features*. Their presence (DS24) and this study's more detailed description of their structure, temporal evolution and dynamics serves to augment and refine the well-established documentation of SPV's overall structure (see Waugh and Polvani, 2010; Schoeberl and Newman, 2015; Butchart, 2022).

The features take the form of transient sub-planetary and synoptic scale, predominantly monopole, positive PV (and relative vorticity) anomalies that translate with and distort the ambient SPV's flow. Their presence can contribute to the





distribution of chemical constituents within the core. Their prevalence at the core's periphery suggests that they can modify the perceived width of the SPV, potentially help mix air both within the core and across its periphery, and serve to diffuse the jet (Bowman and Chen, 1994; Ishioka K., and Yoden, 1995; Mitzu and Yoden, 2001). The latter surmise points to their possible

role, in addition to GWD, in reducing the jet strength at and above the stratopause. Again, the predominantly monopole structure of the features is consistent with their predilection to evolve from a banded structure of enhanced PV near the SPV's periphery and is strongly reminiscent of the growth to finite amplitude of perturbations of a barotropically unstable flow.

This leads to the second assertion that *there is merit in considering the dynamics of a jet-like barotropically unstable flow configuration comprising an annular band of enhanced absolute vorticity to help account for the character of the*

*forementioned observed features.* Our linear instability analysis of such a band indicates that, for a wide range of external parameters characterizing the model's jet structure, the modes of maximum growth possess space-time scales and structure consistent with that of the observed features. In effect barotropic instability provides an attractive mechanism to account for the observed sub-planetary scale features. Contrariwise the model's gravest planetary-scale normal modes ($m = 1, 2$) are shown to be stable and hence do not of themselves support a small amplitude instability hypothesis to account for SSW events.

In addition to the foregoing, our results also bear upon to two other key themes of contemporary SPV studies. First, for mass and chemical composition budget studies and for dynamically studies, it has been customary to identify (- albeit as noted earlier in diverse ways) the SPV's 'edge'. The prevalence of sub planetary-scale features near the SPV's periphery, each associated with a distinctive flow signature, adds further complication to defining the 'edge'. Therefore, it would appear appropriate to characterize the vortex's periphery as often comprising "a rim of finite-width, that is highly variable in both in

space and time, and that corresponds to an annular or fragmented region of enhanced PV".

The study also bears upon the instigation and dynamics of SSW events. Our theoretical results support the conjecture that a vortex breakdown at upper stratospheric levels can result specifically from the synergetic combination of strong planetary-scale Rossby-wave ($m = 1$ or $2$) forcing from below acting upon the vortex's annular PV band. In effect for this model, the annular PV favours the development of patterns somewhat akin to that of SSWs. However, this vortex 'breakdown'

results, not from a displacement or splitting of the SPV's core, but rather from the aggregation of the annular band's PV on the SPV's periphery. The synergetic process is consistent with the concept of a pre-conditioned stratospheric state prevailing ahead of an SSW event. Direct observational corroboration of the conjecture is hampered by the almost ubiquitous presence of large deviations away from symmetric circumpolar flow of the SPV at stratopause levels.

The present study would benefit from more detailed reanalysis-based case studies that include an evaluation of the

role of the parent model's GWD parameterization above the stratopause. Likewise, it would be illuminating to perform numerical simulations with a primitive equation model possessing high horizontal and vertical spatial resolution of potentially unstable basic flow states (- or realized atmospheric states) and undertaken with, and without, strong Rossby wave forcing from below.




*Code and data availability.* The NCAR-NCEP Reanlaysis data was accessed via the NOAA Physical Sciences Laboratory
(http://psl.noaa.gov/). The ERA-Interim data was accessed from the same source, and also directly from the ECMWF data
repositories (Dee et al 2011b). The code for the Contour Dynamics simulation is set out in Dritschel (1989).

*Author contributions.* HCD designed and undertook the observational and theoretical components of the study in consultation
with MAS, and MAS undertook the contour dynamics simulations. HCD wrote the paper, the figures were prepared jointly,
and MAS edited the final version of the text.

*Competing interests.* The contact author has declared that neither of the authors has any competing interests.

*Acknowledgements.* Access to the NCAR-NCEP and ERA-Interim Reanalysis data sets as set out above, is gratefully
acknowledge as is the use of their software. Again, the availability of the original contour dynamics code is appeciated.
*Financial support.* This research has been supported by Prof. Heini Wernli's research group at the Institute of Atmospheric
and Climate Science of the ETH Zurich.

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
