# Peer review of "Transient Flow Patterns of an Annular-like Stratospheric Polar Vortex."

_EGUsphere, 2025_

## Author Comment (AC1)

**Authors response to Reviewer # 1.**

A distinctive feature of the "Weather and Climate Dynamics" journal is its review process. It encourages open collegial and constructive discussion aimed at improving a manuscript's content and further illuminating key issues that relate to the study. We have sought to respond in detail to all the Reviewer's comments in that spirit. Our response is somewhat didactic, and this is prompted by our desire to clarify some possible misconceptions on the part of the Reviewer.

To retain the thread of our response, we first summarize our remarks and then proceed to address in detail the diverse points raised in each of the Reviewer's four comments. For each of the 'four comments' our responses deal with the specific criticisms that are supplement by Appendixes to further clarify the issues.
* * *
**Summary of our Response**
The Reviewer makes two overarching claims regarding the study.

*I:        First, for the observational component, the Reviewer argues that the study is incomplete and that the derived results do not add to our knowledge of the sub-planetary features under discussion.*

The claim of incompleteness falls into the logical category of *"How long is a piece of string?"*. Science being progressive, most research studies could arguably be regarded as incomplete. However, in our manuscript we set out both the overall aim of the study and its specific objectives, and these statements are extracted from the manuscript and listed in our response. Thus, from our standpoint, the study is deemed to be complete. Furthermore, we outline why the Reviewer's stated reason for regarding the study as incomplete is inappropriate.

To counter the claim that no additional knowledge is accrued in the study we present a list of ten of this study's new observationally-based deductions.

It is conceivable that for this component of the study the Reviewer has misconstrued the purpose of the study and perhaps overlooked the new results.

*II:        Second, for the theoretical component, the Reviewer argues there is insufficient reference to, and discussion of, previous studies of barotropic instability in our manuscript, and implies that the derived results for this component might well be neither novel nor substantive.*

The claim of insufficiency is not borne out by inspection of the manuscript. The manuscript's reference list includes no less than 24 papers devoted expressly to this theme and their publication dates range from1962 to 2025. Additionally some of these papers are review articles that consider many additional related papers.

Furthermore, we have at various junctures in the manuscript tersely categorized these previous studies in terms of their strengths, limitations and relevance to the present study, and noted and accounted for when the results in the most pertinent papers accord with, or differ from, the present study's results. In the manuscript we also refer to other papers that consider possible alternative or mixed form of barotropic-baroclinic instability at stratopause levels.

In relation to the claim that the results are neither novel nor substantive, we note that the expressly stipulated goal of the present study was to *not to replicate or refute the results of previous studies but to acquire a deeper understanding of barotropic instability and stability pertaining to the stratosphere.*

Also, we draw attention to key results that are new and substantive and in particular one striking result (- not commented upon by the Reviewer) that is relevant to the occurrence of SSW.

For this component of the study the Reviewer may have not noted the pertinent sections of the manuscript and / or misunderstood the import of the new results.
* * *
**Detailed Responses**

**Reviewer's Comment 1.**

**(a) The Reviewer's overarching objection.**
Arguably the Reviewer's most basic criticism is that our study is incomplete. Such an argument belongs to the logical category of *"How long is a piece of string?"* because it first requires defining what constitutes a complete study of the phenomenon under consideration.

Our overall aim and specific objectives for the study were set out in the manuscript in a sequence of explicit statements (- and are reproduced in Appendix A). Having addressed these issues in our paper, we believe the study to be self-contained and complete. In addition, as is often the custom, we refer in the paper's final Section, to other aspects of the phenomenon under consideration that could merit consideration in future.

In contrast the Reviewer appears focussed on the goal of compiling a climatology of the phenomenon's features. In Appendix B we set out the grounds for regarding such a task, although legitimate in the longer term, to be premature at the present juncture. Moreover, there are even more fundamental objectives that could in principle be formulated for studies of the phenomenon, and that could merit future study, but again these fall outside the scope and objectives of the present study.

(In passing we note the reviewer's criticisms appear to be prompted primarily by our use of the expressions *"only cursory"* and *"limited nature of the observational component of the study"* in the final section. These expressions were introduced only in relation to our **not** aiming to compile a climatology. In effect they are simply a declaration of facts, and do not impinge upon the merit of the study.)

In summary we believe the Reviewer's argument that the study is incomplete because it does not yield (- and was not designed to yield) a climatology is open to dispute.

*(b) The Two Assertions*

Here we address the thrust of the Reviewer's criticisms that relate to the two 'assertions' that were introduced by us in the manuscript's final section to convey and encapsulate the essence of our paper. In that context the Reviewer's objections are essentially a matter of semantics, and on the grounds set out below we discount the accompanying criticisms.

The basis for the dismissal becomes evident on recognizing that in the final section we sequentially noted that:
- the study was not a climatology of the sub-planetary scale features under consideration, and
- the study's results **'lend credence to the two assertions'**.

Thus the Reviewer's objections appear to be based upon either a misreading, a misunderstanding or a misconstruction of the foregoing linked declarations.
In particular we record that :
(i) Nowhere in the paper did we claim our objective was to provide a climatology. As noted above the study's objectives are set out in Appendix A. In effect here the Reviewer appears to castigate us for not undertaking a task that we explicitly said we were not undertaking and that we believe to be premature.

(ii) In this study we did not claim to establish the veracity of the two assertions but rather indicated that our results lent credence to both assertions. Indeed, the study's results are manifestly in accord with the assertions.

To further clarify issues raised directly or tangentially by the Reviewer's objections related to this issue we also supply the following:
-        Appendix B sets out and justifies the rationale underlying the approach pursued in this study, and for us not seeking to compile a climatology at this juncture. This appendix amounts to a vigorous counter to the Reviewer's apparently singular perception of the purpose of the present study and it also serves to negate the basis for the Reviewer's criticism on this point
-        Appendix C provides an extensive list of the numerous new results/deductions made in the observational component of the present study. These items are over and above those made in DS24. In effect the list serves both to counter and to vitiate the Reviewer's implication (here) and declarations (elsewhere) of the non-value of the observational component of the present study.

*(c) Further remarks of the Reviewer in 'Major Comment 1'.*

The Reviewer also further asserts that (i) there are no new conclusions, and (ii) the two assertions are not new.

Point (i): As noted above, we provide in Appendix C a list of the new observational features detected in the present study. (Later in our response we also draw attention to new deductions arrived at in the theoretical component of this study and discuss their import).

Point (ii): Our previous study (DS24) published in 2024 described briefly the sub-planetary scale features detected and noted in the earth's atmosphere. The assertions arose from this recent detection of the features, and it would therefore have been instructive if the Reviewer had substantiated his claims by providing references to the alleged earlier explicit declarations of these assertions.

Other minor points raised by the Reviewer in Comment I, will be discussed in our response to subsequent comments. These refer to (a) the relevance (or otherwise) of studies of other planetary atmospheres, and (b) the stipulated distinctive objectives of, the significance of, and the novelty of, the present study's theoretical results in comparison with that of other earlier studies.
* * *
**Reviewer's Comment 2.**

***(a) Issues related to the establishment of a climatology.***
Here the reviewer makes a series of suggestions / remarks regarding the ingredients and possible composition of a climatology of our detected features. In effect these remarks can be viewed as emphasizing the need to determine the factors that characterize the sub-planetary features. This is indeed the leitmotif of the present study, and the remarks serve admirably to underline the rationale for, and the procedure adopted in, the present study.

Note again that importantly, and contrary to the Reviewer's remark here, we do **NOT** make the assertions attributed to us by the Reviewer. Rather we state explicitly that that our results lend credence to those assertions.

Our study, like other historical case studies of atmospheric flow phenomena, is *per force* confined to examining only a limited number of cases. However, in line with other meteorological case studies, and as stated in the manuscript, the three cases have been selected to be indicative of the SPV's internal structure in fundamentally different flow settings. The presence, scale, structure of the features in each case is notable and thus this study does lay out the necessary groundwork should one wish to establish a climatology.

The Reviewer remarks that there is a need to test an hypothesis regarding the prevalence of the features within / at the edge of the SPV. Routine examination of the day-to-day synoptic charts of the wintertime SPV at stratopause levels in both hemispheres readily confirms that on an almost daily basis. However, like their counterpart in the troposphere, their day-to-day appearance exhibits a range of complex patterns, as one might expect from a chaotic system, and this will render the compilation of a climatology demanding but not insuperble. We have previously undertaken and published a study (referenced in our manuscript) of related synoptic PV features in the lowermost polar stratosphere.

***(b) The Reviewer's issue with the vertical extent of the detected features.***
The Reviewer stresses several times in this section, and elsewhere, that in the context of compiling a climatology, it is the desirable to assess the vertical coherence of the detected features. We agree.

However more generally we recognize, that from a dynamical standpoint, it is also desirable to determine the location of their origin and the nature of their decay, their spatial and temporal scale, their thermal and circulation signatures, their balanced or unbalanced state, the extent of their conservation of PV during their life cycle, their relationship to upward propagating Rossby and inertio-gravity waves, their intra-seasonal character, and their relationship to the distribution of chemical constituents.

An overt aim of the study is to draw attention to these issues, and our study was predicated upon the desirability of shedding some light upon all these characteristics, and our diagnosis and accompanying figures were geared to at least partially meet these multiple aims. With regard to the vertical coherence of the features, we note here that in the three cases considered the features extended over at least a scale height below the extant stratopause (- and likely to extend over a somewhat similar height above the stratopause). This was evident in the material already displayed for two of the three cases.
* * *
**Reviewer's Comment 3**

**(a) The perceived lack of reference to and discussion of previous observational studies.** The Reviewer argues that there is a need to reference other previous related studies. This claim of insufficiency is not borne out by inspection of the manuscript. It includes references to approximately 70 papers (24 to the theme of barotropic instability) and each one bears upon the paper's objectives. Moreover, many of the referenced papers are Review Articles devoted to particular themes that are central to the study and that abound with additional references.

Notwithstanding we comment here on the relevance (or otherwise) of papers that the Reviewer urges us to discuss.

(a) The first is that of *Harvey et al (2009)* and the reviewer states unequivocally that in this study similar features have been observed to those noted in DS24. Clearly if the sub-planetary features are endemic in the wintertime at stratopause elevations then it is likely that a chart incorporating data of the required resolution might well / will capture such features. Indeed, on close inspection we have been able to detect vestiges of these features in some of the earlier classic papers, but significantly no direct comment has been made regarding their presence. (This fact was already noted in passing in DS24 but has not been repeated in the present paper.)

The study of *Harvey et al* falls into this category. The only pertinent diagrams in the paper are two panels of their Fig. 1, but the paper's authors remark only that in one panel there is an isolated maximum PV centred over the pole (not unexpected) and that in the other there in a minimum of PV in another region. No specific reference is made to the presence of sub-planetary scale features although inspection shows that there are two or three such features (again consistent with our expectation and other early studies). In the current context it is noteworthy that a more pertinent paper is that of *Harvey et al 2002* and this paper is already referenced in our manuscript. It is conceivable that the Reviewer had not noted this study.

(b) The second study (*Hughes et al 2025*) was published almost concurrently with the present study and was not accessible to us prior to our own submission. The Reviewer indicates that *"the article is a good place to start for this literature"*.

We presume that the inference of this suggestion is that our study is deemed not to be sufficiently *au fait* with the pertinent literature or that *Hughes et al 2025* provides the necessary description of the current state-of-the-art in the field. In response we note the following:-

(i) The *Hughes et al 2025* article is concerned principally with vortices in the Martian atmosphere and might not be ideal for the present study. For example it does (a) not reference the key paper for the present study (DS24) and which would help link it to the earth's atmosphere, and (b) essentially precludes direct consideration of SSW events in the earth's stratosphere that forms an integral part of the present study,

(ii) in terms of the up-to-date 'currency' of our study we note that the most recent paper referenced (- albeit incorrectly) in *Hughes et al* is already listed in the reference list of our study, along with several recent papers and review articles that relate to barotropic instability in other planetary atmospheres,

(iii) trenchantly, the central thrust of the *Hughes et al* article is, as noted above, on the Martian atmosphere. To be relevant to barotropic stability considerations for the earth's atmosphere and more pointedly to this study, it would be necessary for the the key non-dimensional parameters for the Martian atmosphere to be at least of the same order of magnitude as that for the earth's atmosphere. The Reviewer is invited to consult, for example, the study of *Michaelangeli et al. (1987)* to verify whether this is the case.

In summary we suggest that, it could be justifiably argued the Reviewer's assertion that the *Hughes et al* article *"is a good place to start"* should not be taken at face value.

The third paper (*Serviour et al 2017*) that the Reviewer urges us to discuss is primarily a theoretical and numerical model study. The strategy adopted therein is akin to that pursued in the present study but uses the divergent shallow water equations (SWEs). We assume that the Reviewer's previous queries regarding the vertical coherency of the observed features and the vertical structure of the SPV is linked to this reference to the *Serviour et al.* study. The relative performance of the SWE relative to a non-divergent equivalent is set in Paldor et al, but as we note in our final section the key arbiter would be a comparison of idealized models with the results of a multi-level PE model. Strikingly, the results of our study can provide an observationally-based indication of the real-world relevance of say a 'divergent SWE' and the classical 'equivalent barotropic' approaches to the evolution of the features under consideration.
* * *
**Reviewer's Comment 4 :**

**Criticisms of the study's theoretical component**

Here the Reviewer requests that more attention be given to previous studies on the topic of barotropic instability, questions the value of the theoretical component of our study, queries

whether the results are new, and argues that there is insufficient warrant for including the details of our linear theoretical analysis.

To counter these remarks we record below the strategy, objective and treatment of the above issues in our manuscript, set out the nature and novelty of the results, and indicate the manner in which they reflect crucially upon, and refute or rebut, the above criticisms of the Reviewer.

**(A) The absolutely key paragraph** in our manuscript regarding this component of the study is the following (Lines 71-74)
*The strategy adopted here is to select an idealized theoretical model capable of capturing the essence of rapidly growing wave patterns of a barotropic model for flow settings akin to that of a deep undisturbed SPV. The objective is not to replicate (or contrast) the results with that of the earlier studies but rather to seek further basic understanding of and insight upon the dynamics of the observed features.*

These declarations are in direct contrast to the Reviewer's presumptions regarding the purpose of our study and the linkage to previous research.

**(B)** In relation to previous research, we noted that  (Lines 68-71)
*The ready inference that an SPV's jet can satisfy a necessary condition for barotropic instability has spawned a fleet of studies for basic states akin to that of either the SPV's jet or the circumpolar jets of other planetary atmospheres (see for example, Pfister, 1979;  Davies, 1981; Hartmann, 1983; Michaelangeli et al., 1987; Manney et al., 1988; Seviour  et al 2017; Mitchell et al., 2021; Waugh et al., 2023).*

This is a list of the most pertinent papers.

**(C)** Likewise (Lines 347-349)
*Our objective here is to explore the character of perturbations that can evolve on an annular-like flow structure akin to that of a deep, axisymmetric and undisturbed SPV. To this end we parsimoniously adopt an idealized theoretical model capable of representing fundamental features of such a setting including the occurrence of barotropic instability.*

In effect we have set out here the rationale for adopting our particular model

(D) and we cautioned explicitly that (lines 351-354)
*....it excludes consideration of three-dimensional effects and therefore can only represent flows structures that are comparatively deep. These restrictions, together with the model's simple configuration, imply that our results will constitute a potentially insightful but qualitative guide rather than providing quantitative definitive statements.*

This is a direct description of the adopted model's limitations and the manner in which the results should be viewed.

**(E)** With regard to the results of the linear instability theory we recorded that (Line 410-413, and 486-487).

*Results are derived for the linear and non-linear response of perturbations of stipulated basic states. The linear component includes a normal mode analysis that establishes the growth rate and azimuthal wavenumber (m) of unstable modes, and an analysis of the response to a specific form of non-normal mode initial perturbation.*

And
*The general dependency of the instability upon the configuration of the basic state is illustrated in Fig. 13. It shows the growth rate (σ) as a function of the wavenumber (m) for the settings corresponding to Fig.12.*

Here we have indicated the form in which the results can be readily presented. Many, if not most, previous studies could not provide this comprehensive information in this direct form.

**(F)** With regard to the contour integration technique we noted (Line 413-415)
*with adjustments of the standard contour advection technique to (a) account for the specification of the absolute as opposed to the relative vorticity and (b) circumvent the standard contour advection stipulation of non-zero absolute vorticity in the outermost domain.*

These again are novel features of our study,

**(G)** The nature of the derived results is set out as follows (Lines 486- 493)
*The growth rate increases markedly with jet strength, decreases strongly with the width of the annulus, and is comparatively insensitive to the radial velocity decay of in the outer region. In effect the sensitivity to jet skewness is primarily governed by the structure within the core. These results are in qualitative agreement with, and complement those of, earlier barotropic instability studies (see in particular Hartman, 1983) but with some differences for the graver modes (- see later). The large growth rate of the sub-planetary modes is notable since a pragmatic lower bound for an unstable mode's growth rate would be that σ should exceed t, where t ~ (10 days)$^{-1}$ is the radiative decay time-scale (Shine, 1987) of deep perturbations within the polar-night vortex in the upper stratosphere.*

Thus, our derived results represent in a terse form the thrust of previous results and even more notably highlight key and substantive differences - we account for these differences below in a later quotation

**(H)** The application of the PV perspective to derive the Ready Reckner and interpret the results (Lines 496-592)
*To pinpoint the dependencies illustrated by, and dynamics underlying, the results displayed in Fig. 13, we resort to the PV-based phenomenological interpretation (Lighthill, 1967; Hoskins et al 1985) and mathematical formulation (Davies & Bishop, 1994) of quasi-geostrophic instability. In this framework the present instability is viewed, in terms of the two waves propagating respectively on the opposite-signed PV jumps at r=a and r=b , and held stationary relative to one another by the differing in-situ flow at their in-situ locations. In such a configuration the waves inter-dependency can promote mutual amplification and modification of their azimuthal phase velocities.*

This approach is essentially novel to considering barotropic instability in non-Cartesian flow systems.

(I) The significance and simplicity of the instability's interpretation. (Lines 546-548).
*In effect the Ready Reckoner is consistent with, and provides a compact account of, the results displayed in Fig. 13. A further deduction is that, in conformity with the cases study examples of Section 2, sub-planetary and synoptic-scale features can evolve preferentially for a wide-range of realistic annular flow settings.*

The entire instability results for the most unstable modes wavenumber and growth rate reduces to the trivial evaluation of Eqs (10) and (11). We personally regard this as a delightful result that is useful, novel and insightful result.

**(J)** The character/stability of the graver modes (Section 3.2.4).
We view this result to be seminal, runs counter to that recorded in previous studies, is consistent with our dynamical interpretation, and is accounted for later in the manuscript.

**(K)** Non-linear evolution of linearly unstable modes. (Section 3.4)
The simplicity of our adopted model enables us to determine the full spectrum of unstable modes (- this is hardly ever available in other SPV instability studies) and the non-linear integrations confirm that the most unstable modes are the ones that are preferentially sustained in the non-linear phase. This confirmation is not available for many studies. Interestingly, for a somewhat unrealistic basic flow settings for the earth's stratosphere, it is shown that a radically different form of evolution can transpire. This result is novel and might have implications for other planetary atmospheres.

**(L)** The limitations of this and other previous studies         (Lines 679-689)
*The theoretical component of the study is constrained by the adoption of a non-divergent barotropic model. Positively the model does not need to invoke quasi-geostrophy (c.f. Davies, 1981; Hartmann, 1983), incorporates an acceptable representation of the b-effect (c.f. Dritschel, 1989; Dritschel and Polvani, 1992) as opposed to spherical geometry (c.f. Hartman,1983; Michaelangeli et al.,1987; Manney et al., 1988; Mitchell et al., 2021; Waugh et al., 2023), and eminently enables the derivation and interpretation of analytical solutions for a wide range of relatively realistic basic states velocity fields. Negatively it excludes consideration of baroclinic (c.f. Dickinson, 1983; Simmons 1974) and mixed barotropic-baroclinic (c.f. Manney & Randel, 1993) instability and is best at representing barotropic states and deep perturbations (Paldor, et al 2021). The character of the observed features signals that these shortcomings are not pathological.*

We believe this to be a compact description of this and other studies limitations.

**(M)** The Reviewer's request that we discuss certain other studies can be considered in the context of the response in point **(L)**. We note that :
- *Dritschel, D. G., 1986:* The initial configuration does not relate directly to, nor is relevant to, the present study.
- *Dristchel, and L. M. Polvani, 1992*: This study pertains to non-rotating atmospheres. There is another paper of *Dritschel et al* that is more relevant and that is related to this study and

that is referenced by us. Our model would reduce to that considered in that study in the limit of a non-rotating atmosphere but would also be for somewhat unrealistic basic state azimuthal wind profiles.

The following papers :
*Hughes et al 2025*; *Harvey, V.L., Randall, C.E. and Hitchman, M.H 2009*; *Michelangeli, D. V., R. W. Zurek, and L. S. Elson, 1987*; are noted elsewhere in our response.

The study of *Rozo et a*l does not appear o relate to the present study.

**(N)** A There is an interesting side remark that we have drawn attention to in some of our earlier responses (Lines 687-689).
*Here the normal mode graver modes (m = 1, 2) are stable, and contrast with the instability of some earlier studies undertaken with a special basic state structure (Hartmann, 1983). This discrepancy is attributable to a second, and generally unrealized, mid-latitude PV annulus in the special formulation.*

 The implication here is that the results of several of the classic studies referenced in this paper need to be viewed with some caution. To our knowledge this important caveat is not necessarily generally recognized.

**(O)** The non-normal mode behaviour of a perturbed annulus.
Finally, we draw attention to a major theme of the study, and an extract of its conclusion reads  (Lines 718- 722)
*In effect for this model, the annular PV favours the development of patterns somewhat akin to that of SSWs. However, this vortex 'breakdown' results, not from a displacement or splitting of the SPV's core, but rather from the aggregation of the annular band's PV on the SPV's periphery. The synergetic process is consistent with the concept of a pre-conditioned stratospheric state prevailing ahead of an SSW event.*

We regard this as to be a new, fundamental and of potentially high significance for stratospheric dynamics. (There is no indication that the Reviewer has noted the nature and significance of this result.
* * *
Together the forgoing detailed 15 responses (A-O) to the Reviewer's FOURTH comment stand clearly in stark contrast to the Reviewer's criticisms, and we believe negate their implications.
* * *
**APPENDIX A** *: The study's structure, aims and objectives.*
The stipulated structure, aims and objectives of the present study were set out explicitly in the manuscript in a sequence of statements. These that are repeated below.

**Line (29)**       "The present study is geared to examining the planetary and finer-scaled flow features that are contiguous to, or related to, the structure of the SPV's periphery,"

**Line (62)**      In this study attention is focused on the presence, role and dynamics of a PV annulus and/or its fragmented counterpart in the wintertime upper stratosphere. It is set against the backcloth of the series of questions posed above, ….      For each case attention is drawn to the structure, temporal evolution, and dynamical character of the realized transient sub-planetary and planetary scale flow features.

**Line (71)**      The strategy adopted here is to select an idealized theoretical model capable of capturing the essence of rapidly growing wave patterns of a barotropic model for flow settings akin to that of a deep undisturbed SPV. The objective is not to replicate (or contrast) the results with that of the earlier studies but rather to seek further basic understanding of and insight upon the dynamics of the observed features.

**Line (90)**      Three cursory case-studies are presented with the objective of capturing the essence of transient flow features associated with an annular-like PV structure in the wintertime upper stratosphere.

**Line (106)**      The objective is to establish the existence, scale, structure, evolution and dynamics of the sub-planetary transient flow features.
* * *
*APPENDIX B: On the issue deriving a Climatology.*

It is evident from the content of the manuscript (see APPENDIX A) that the purpose of the present study was NOT to compile a climatology. It is in that context that our bald statements *"only cursory"* and *"limited to the three case studies and hence can merely hint at the climatology"* is simply a declaration of that fact.

        Prior to compiling a climatology of a phenomenon it is highly desirable, indeed requisite, to establish the essence or major characteristics of that phenomenon such as its scale, structure, dynamics, etc. Given the highly limited prior information available regarding these features (i.e. the contents of the short note DS24) it is clear that additional information needs to be acquired ahead of establishing a climatology of the features.
        One facet of the present study is that it prepares the groundwork for such a task, and we believe that it consitutes a reasonable and necessary first step. (Put metaphorically in terms of a Southern US proverb -  *'to make bear soup one must first catch a bear'*).

In summary the Reviewer states and believes that we should have prepared a climatolgy. We disagree, and our paper can be viewed as pinpointing some of the characteristics of the features that are basic for preparing a climatology. Indeed some of the Reviewer's other comments are in line with our own opinion.
* * *
*Appendix C:  A list of some of the new observationally diagnosed results.*

The Reviewer is of the opinion that, over and above the results recorded in DS24, the observational component of the present study does not include new aspects related to the sub-planetary features or that add significantly to the content of the earlier short note.

Below is a list of new aspects of these features that are explored and recorded in our study and that are deemed to be of seminal importance.

(i)     consideration of the features in three fundamentally different flow settings,

(ii)    insight on the relationship of the features with planetary scale Rossby waves propagating toward the stratopause from lower elevations,

(iii)   identification of the vertical coherence of the features as evidenced by the PV and relative vorticity patterns at a range of levels,

(iv)    scale and amplitude of the divergence patterns at stratopause levels in the presence of the features, and the implications for the nature of the associated dynamical balance,

(v)     depiction of, and deductions related to, the specific humidity pattern evident in the ERA-Interim analysis,

(vi)    a detailed portrayal of the space-time development of the PV pattern of the features on key isentropic surfaces, and consideration of the attendant implications for their translation, dvelopment and PV conservation,

(vii)   a more detailed inter-hemispheric wintertime comparison,

(viii)  the distinctive signature of the potential temperature field related to the features,

(ix)    the even more distingtive associated signature of the ozone mass mixing ratio,

(x)     the break up of the annular band of enhanced PV ahead of, during, and after an SSW as evident in different levels, and the novel implications for the  development of an SSW,

This partial list of the new insights clearly does not concurr with the Reviewer's claims.